# A library of lineage-specific driver lines connects developing neuronal circuits to behavior in the *Drosophila* ventral nerve cord

Jelly HM Soffers[1]*, Erin Beck[1], Daniel J Sytkowski[1], Marianne E Maughan[1], Devasri Devarakonda[1], Yi Zhu[2], Beth A Wilson[2], Yu-Chieh David Chen[3], Ted Erclik[4,5], James W Truman[6], James B Skeath[2], Haluk Lacin[1]*

[1]School of Science and Engineering, Division Biological and Biomedical Systems, University of Missouri-Kansas City, Kansas City, United States; [2]Department of Genetics, Washington University School of Medicine, St. Louis, United States; [3]Department of Biology, New York University, New York, United States; [4]Department of Biology, University of Toronto - Mississauga, Mississauga, Canada; [5]Department of Cell and Systems Biology, University of Toronto - Mississauga, Mississauga, Canada; [6]Department of Biology, University of Washington, Seattle, United States

**\*For correspondence:**
j.soffers@umkc.edu (JHMS);
haluklacin@umkc.edu (HL)

**Competing interest:** The authors declare that no competing interests exist.

## eLife Assessment

This work presents an **important** genetic toolkit for *Drosophila* neurobiologists to access and manipulate neuronal lineages during development and adulthood. The evidence supporting the fidelity of this toolkit after revision is **compelling**. This work will interest *Drosophila* neurobiologists in general, and some of the genetic tools may be used outside the nervous system. The conceptual approaches used in this paper are likely transferable to other fields as comparable data and genomic methods are obtained.

**Abstract** Understanding developmental changes in neuronal lineages is crucial to elucidate how they assemble into functional neural networks. Studies investigating nervous system development in model systems have only focused on select regions of the CNS due to the limited availability of genetic drivers that target specific neuronal lineages throughout development and adult life. This has hindered our understanding of how distinct neuronal lineages interconnect to form neuronal circuits during development. Here, we present a split-GAL4 library composed of genetic driver lines, which we generated via editing the genomic locus of lineage-specific transcription factors and demonstrate that we can use this library to specifically target most individual neuronal hemilineages in the *Drosophila* ventral nerve cord (VNC) throughout development and into adulthood. Using these genetic driver lines, we found striking morphological changes in neuronal processes within a lineage during metamorphosis. We also demonstrated how neurochemical features of neuronal classes can be quickly assessed. Lastly, we documented behaviors elicited in response to optogenetic activation of individual neuronal lineages and generated a comprehensive lineage-behavior map of the entire fly VNC. Looking forward, this lineage-specific split-GAL4 driver library will provide the genetic tools needed to address the questions emerging from the analysis of the recent VNC connectome and transcriptome datasets.

## Introduction

Neuronal circuits underlie nervous system functions ranging from perception and movement to cognition and emotion. Most neurons found in the adult CNS of animals are generated and assembled into circuits during development. Investigating the formation of these circuits provides valuable insights into the functional organization and operation of the nervous system, both in health and disease.

*Drosophila* has served as a powerful model system to investigate how neuronal circuits function due to its medium complexity compared to vertebrate models yet rich repertoire of behaviors and unprecedented genetic toolkit. High-resolution electron microscopy data of the adult fly brain and ventral nerve cord (VNC) reveal individual neuronal morphologies and their synaptic connections (*Marin et al., 2024*; *Azevedo et al., 2024*; *Li et al., 2020*; *Scheffer et al., 2020*; *Schlegel et al., 2023*). The integration of these morphological data with single-cell transcriptome profiles has placed the adult fly CNS at the forefront of studies of circuit operations at the molecular level (*Allen et al., 2020*; *Bates et al., 2019*; *Özel et al., 2021*; *Yoo et al., 2023*).

In *Drosophila* and other model systems, less attention has been paid to how neuronal circuits develop compared to how they function, limiting our understanding of the developmental processes that instruct newly born neurons to assemble into functional circuits. In *Drosophila*, the same set of neural stem cells, called neuroblasts (NB), sequentially form the larval and adult CNS, with the adult CNS having 10–20-fold more neurons and greater complexity. Some of the embryonic-born neurons, which function in the larval CNS, are remodeled to integrate into adult circuits (*Prokop and Technau, 1991*; *Truman, 1990*; *Truman and Bate, 1988*). Most of the adult neurons are born post-embryonically during larval and early pupal stages. These neurons fully differentiate and assemble into circuits during metamorphosis into the adult, which lasts several days. This extended window of neurogenesis and neuronal maturation during the formation of the adult VNC facilitates experimental manipulations that are not feasible during the brief period of neurogenesis in the embryo, such as temporal gene silencing studies to discriminate axon guidance and synapse formation.

The fly VNC, like its vertebrate analog, the spinal cord, is functionally compartmentalized into lineally related groups of neurons, called neuronal lineages. In flies, Notch-mediated asymmetric cell division divides the neuronal population of each NB into two subclasses, called hemilineages: 'A' hemilineages are composed of Notch ON cells and 'B' hemilineages are composed of notch OFF cells (*Skeath and Doe, 1998*; *Spana and Doe, 1996*; *Truman et al., 2010*). The adult fly VNC is composed of ~15,000 neurons, most of which are found in the three thoracic segments. Each thoracic hemisegment contains 34 major post-embryonic hemilineages, with some segment-specific variation in the type of hemilineages and their morphology. Recent studies identified these hemilineages in the VNC Electron Microscopy (EM) volume dataset and showed that neurons within a given hemilineage exhibit a stereotyped pattern of connectivity (*Marin et al., 2024*; *Azevedo et al., 2024*; *Ehrhardt et al., 2023*; *Lesser et al., 2024*). This revealed that hemilineages display a propensity to form synaptic connections with neurons from other specific hemilineages, uncovering a macro-connectivity among hemilineages. Hemilineage-based compartmentalization of the VNC is also observed at the level of gene expression. (*Allen et al., 2020*) assessed the transcriptome of the entire adult VNC via single-cell RNA sequencing (scRNAseq) and showed that hemilineage identity correlates highly with unique clusters of cells, which are partitioned solely based on gene expression via dimensionality reduction. Lastly, several studies employing hemilineage-restricted neuronal manipulations showed that the VNC hemilineages represent functional modules that control animal behavior (*Agrawal et al., 2020*; *Harris et al., 2015*; *Lacin et al., 2020*). Indeed, like the cardinal classes of interneurons in the spinal cord (*Briscoe et al., 2000*; *Jessell, 2000*; *Lu et al., 2015*), hemilineages in the *Drosophila* VNC are functional units, each contributing to aspects that control specific behaviors. Thus, taking a hemilineage-based approach is essential for a systematic and comprehensive understanding of behavioral circuit assembly during development in complex nervous systems.

Addressing the question of how neurons in individual hemilineages develop into functional circuits requires genetic tools to manipulate individual hemilineages throughout development. Existing genetic driver lines (GAL4, Split-GAL4, and LexA libraries) are often limited in their use for developmental studies. The expression of such drivers typically relies on 2–3 kb genomic fragments that contain enhancer elements, *Meissner et al., 2024* and comprehensive screening efforts have identified driver lines that mark specific neuronal populations (*Meissner et al., 2024* and citations therein). However, the genomic fragments driving GAL4, Split-GAL4, or LexA expression lack the complete

endogenous transcriptional control mechanisms. As a result, they are oftentimes only expressed in specific neuronal populations during particular life stages, such as exclusively in the larva or adult. Consequently, they lack the temporal stability required for comprehensive developmental analysis (*Meissner et al., 2024*; *Luan et al., 2020*; *Pfeiffer et al., 2010*), highlighting a critical need for developmentally stable and hemilineage-specific driver lines. These tools will allow us to track and measure individual hemilineages as well as activate or inactivate specific genes and neuronal functions within them, thereby facilitating the identification of fundamental principles underlying circuit development.

Here, we describe a split-GAL4 library that targets unique hemilineages in a developmentally stable manner. Our previous work demonstrated that many of the lineage-specific transcription factors that regulate the specification and differentiation of post-embryonic neurons are stably expressed during development (*Lacin et al., 2014*; *Lacin et al., 2019*; *Lacin and Truman, 2016*; *Lacin et al., 2024*). We employed gene knock-in strategies to insert split-GAL4 driver coding sequences in frame with the coding regions of these transcription factors, generating lineage-specific, temporally stable driver lines for selected hemilineages (*Lacin et al., 2020*; *Lacin et al., 2014*; *Lacin et al., 2019*). To extend this approach to all hemilineages in the VNC, we required a more comprehensive list of lineage-specific transcription factors. To compile this list, we utilized published scRNA-seq data of the VNC and expanded upon the work of *Allen et al., 2020*; *Lacin et al., 2014*; *Harris et al., 2015*. This work had assigned a part of the scRNAseq cell clusters to hemilineages. We analyzed the gene expression patterns of the remaining clusters for combinations of significantly enriched transcription factors, referred to as cluster markers, and tested the expression patterns of these genes with genetic reporter lines and antibody staining. We were able to assign 33 of the 34 major hemilineages to scRNAseq clusters with this approach. Then, we generated gene-specific split-GAL4 lines for 28 of these hemilineage-specific transcription factors via genome editing and recombination techniques. We performed a thorough analysis of the expression patterns of binary combinations of split-GAL4 AD and split-GAL4 DBD lines using combinations of the 28 transcription factor-specific hemidrivers we present in this study and split-GAL4 lines generated previously (*Lacin et al., 2019*; *Lacin et al., 2024*; *Chen et al., 2023a*). We report 44 combinations that target 32 of the 34 VNC hemilineages; most of these drivers do this specifically and in a developmentally stable manner. Finally, we demonstrate the ability of this library to map neurotransmitter expression to individual hemilineages and to map specific behaviors to defined neuronal lineages.

## Results

### Intersecting the expression of *acj6* and *unc-4* with the split-GAL4 method faithfully marks hemilineage 23B throughout development and adult life

Many transcription factors in the CNS are expressed in a hemilineage-specific manner, and their expression is generally maintained throughout the lifetime of the neurons that express them (*Lacin et al., 2014*). We asked whether we could generate specific and temporally stable driver lines by hijacking the expression of such transcription factors. We initially focused on Acj6 and Unc-4, which are transcription factors expressed in numerous neuronal cell clusters in the brain and VNC (*Figure 1A–B*). Our prior work demonstrated that these proteins are co-expressed exclusively in hemilineage 23B neurons in both the larva and adult (*Lacin et al., 2014*). We leveraged this unique co-expression pattern to develop a genetic set-up that targets only the 23B neurons in a developmentally stable manner. We combined two techniques: the Trojan-exon-based driver for target gene transcription (*Diao et al., 2015*) and the split-GAL4 method (*Luan et al., 2006*). The split-GAL4 method works by reconstituting GAL4 function through the interaction of GAL4's DNA-binding domain (DBD) and an activation domain (AD) in cells where both transgenes are expressed. Here, we used the *unc-4* split-GAL4 AD and DBD lines that we had previously generated (*Lacin et al., 2020*) and created *acj6* split-GAL4 lines by replacing the MIMIC insertion in the *acj6* coding intron with a Trojan exon encoding either p65.AD or GAL4-DBD via recombinase-mediated cassette exchange (RMCE).

By combining *unc-4*-GAL4^AD and *acj6*-GAL4^DBD transgenes in the same animal with a nuclear expression reporter gene, UAS-nls-tdTomato, we specifically visualized the thoracic clusters of 23B neurons in the adult CNS (*Figure 1C*). Small clusters of neurons were evident in the subesophageal zone, and their projections suggest that they are the labial homologs of hemilineage 23B (*Figure 1—figure*

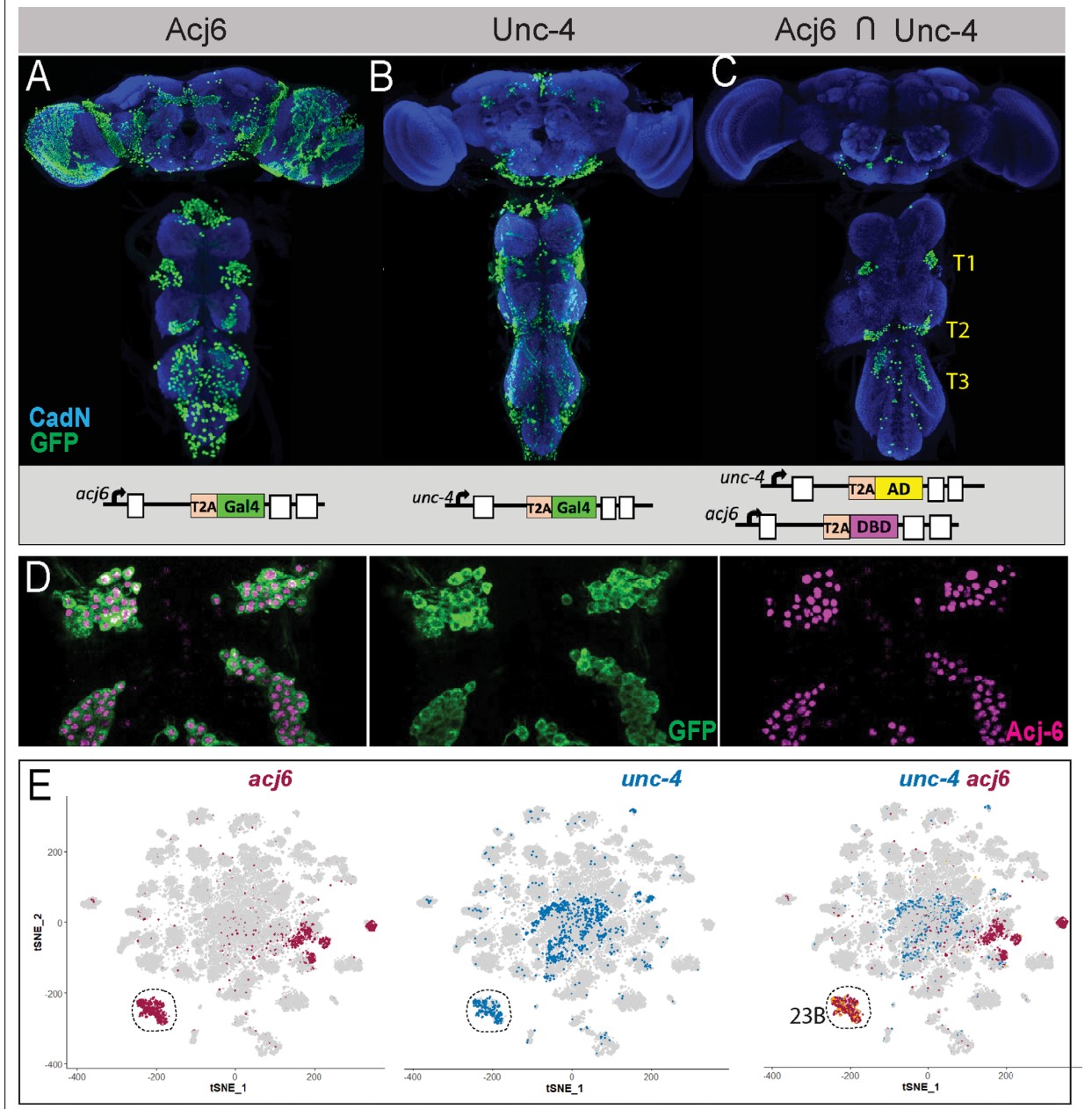

**Figure 1.** Intersecting the expression of *acj6* and *unc-4* genes with the split-GAL4 method faithfully marks hemilineage 23B. (**A–C**) Projections of confocal stacks of the adult VNC. Blue: CadN; (**A**) *acj6*-GAL4 driven nls-tdTomato expression (displayed in green) marks Acj6 expressing neurons. (**B**) *unc-4*-GAL4 driven nls-tdTomato expression (displayed in green) marks Unc-4 expressing neurons. (**C**) The intersection of acj6 and unc-4 expression (displayed in green) (*acj6*-GAL4$^{AD}$, *unc-4*-GAL4$^{DBD}$> UAS-nls-tdTomato) marks lineage 23B neurons in the SEZ and VNC. (**D**) A partial confocal projection showing the complete overlap between membranous GFP (green) and Acj6 (magenta) immunostainings in *acj6*-GAL4$^{AD}$, *unc-4*-GAL4$^{DBD}$-marked 23B neurons in the adult VNC (T1 and T2 segments shown). (**E**) scRNAseq t-SNE plot shows Acj6 (Purple) and Unc-4 (Dark Blue) co-expression in a group of cell clusters.

The online version of this article includes the following figure supplement(s) for figure 1:

**Figure supplement 1.** *acj6*-GAL4$^{AD}$, *unc-4*-GAL4$^{DBD}$-driven myr-GFP marks 23B neurons throughout development.

*supplement 1D, E*). Membranous GFP expression (UAS-myr-GFP) also highlighted axonal projections of a few leg, gustatory, and antennal sensory neurons which are missed with nuclear-based methods such as immunostaining for nuclear transcription factors or nuclear GFP reporter genes, since sensory cell bodies are located outside of the CNS. The reverse combination (*unc-4*-GAL4^DBD and *acj6*-GAL4^AD) exhibited an almost identical expression pattern (not shown).

To verify that these gene-specific split-GAL4 drivers recapitulate the intersected expression patterns of *unc-4* and *acj6*, we performed immunostaining with antibodies against Acj6 and Unc-4 on embryos carrying the described transgenes and evaluated the overlap with the GFP signal. Robust GFP expression was observed in the late-stage embryo and marked segmentally repeated clusters of neurons in the VNC (*Figure 1—figure supplement 1A, B*). All GFP-positive cells were also positive for Acj6 and Unc-4 immunostaining, indicating that these cells correspond to the embryonic progeny of NB7-4, embryonic 23B neurons (*Lacin et al., 2020*). Occasionally, one to two cells per segment expressed both transcription factors but not GFP (not shown). These cells, located ventrally, are likely late-born, immature neurons and their GFP expression may lag endogenous gene expression of Acj6 and Unc-4 due to the additional round of transcription and translation required for GFP expression. Outside the CNS, GFP-positive sensory neurons were found in the embryonic head region, where taste organs are located (not shown). Overall, the embryonic expression analysis confirmed that the *acj6*-GAL4^DBD and *unc-4*-GAL4^AD split-GAL4 combination accurately recapitulates co-expression of Acj6 and Unc-4 proteins and target embryonic 23B neurons. To test whether this combination of split-GAL4 driver lines also specifically marks the 23B hemilineage during larval, pupal, and adult life, we carried out similar analysis during these stages. Like in the embryo, the intersection of *acj6*-GAL4^DBD and *unc-4*-GAL4^AD specifically marked 23B neurons in the larva, pupa, and adult (*Figure 1—figure supplement 1C, D*). Thus, this split-GAL4 combination effectively targets reporter expression specifically to the 23B neurons in the VNC from the early larva through to the adult.

## Identifying new marker genes for hemilineages and assigning hemilineages to scRNAseq clusters of the VNC transcriptome

The example described above demonstrated that combining the Trojan exon method with the split-GAL4 approach holds the potential to generate temporally stable, lineage-specific driver lines for every hemilineage in the VNC, provided suitable pairs of genes are identified. Our prior work created a map of the expression of 20 transcription factors, each of which is expressed from early larval stages to the adult in most or all neurons of a small number of hemilineages in the adult VNC (*Lacin et al., 2020*; *Lacin et al., 2014*; *Lacin et al., 2019*). When overlapped in a binary manner with each other, these transcription factors uniquely identify more than half of the 34 adult VNC hemilineages, rendering them ideal genomic targets from which to create a library of split-GAL4 driver lines.

To identify unique binary gene combinations that can selectively label each of the remaining hemilineages, we further analyzed scRNAseq data from the adult VNC (*Allen et al., 2020*). Allen et al. defined 120 t-SNE clusters based on unique combinations of significantly enriched genes, referred to as cluster markers. By comparing these cluster markers to established lineage markers, the Goodwin group assigned 18 hemilineages to one or more clusters, leaving 16 hemilineages unassigned. For example, they assigned grouped clusters 67, 93, 35, and 51 to lineage 23B. In agreement with our immunostaining that revealed that cluster markers *acj6* and *unc-4* mark this hemilineage (*Figure 1C and D*), we report that also the expression patterns of *acj6* and *unc-4* expression overlap in this grouped scRNAseq cluster (*Figure 1E*). We continued this approach and tested whether other cluster-specific marker genes were expressed in their corresponding hemilineages. For instance, Allen et al., assigned clusters 0 and 100 to hemilineage 4B. Both clusters express *fkh, HLH4C,* and *oc* genes in addition to three additional genes: *hb9* (also known as *exex*), *HGTX*, and *ap* which we had previously shown to be expressed in 4B neurons (*Lacin et al., 2009*). Using GFP-tagged BAC reporter lines for *fkh, oc,* and *HLH4C* combined with immunostaining for Hb9, we demonstrate that cluster markers *fkh, oc, and HLH4C* are indeed expressed in 4B neurons in the larval and adult VNC, consistent with the scRNAseq data (*Figure 2A*, data not shown). In addition to hemilineage 4B, Hb9 marks hemilineage 10B and 16B neurons (*Kuert et al., 2014*). Hemilineage 10B was assigned to clusters 39, 68, and 91 and hemilineage 16B to clusters 5 and 46 (*Allen et al., 2020*). Knot (Kn) is a marker for clusters 39 and 91, and Sp1 for clusters 5 and 46. Reporters for both genes show that Kn and Sp1 are expressed in lineage 10B and 16B neurons, respectively (*Figure 2B and C*). Therefore,

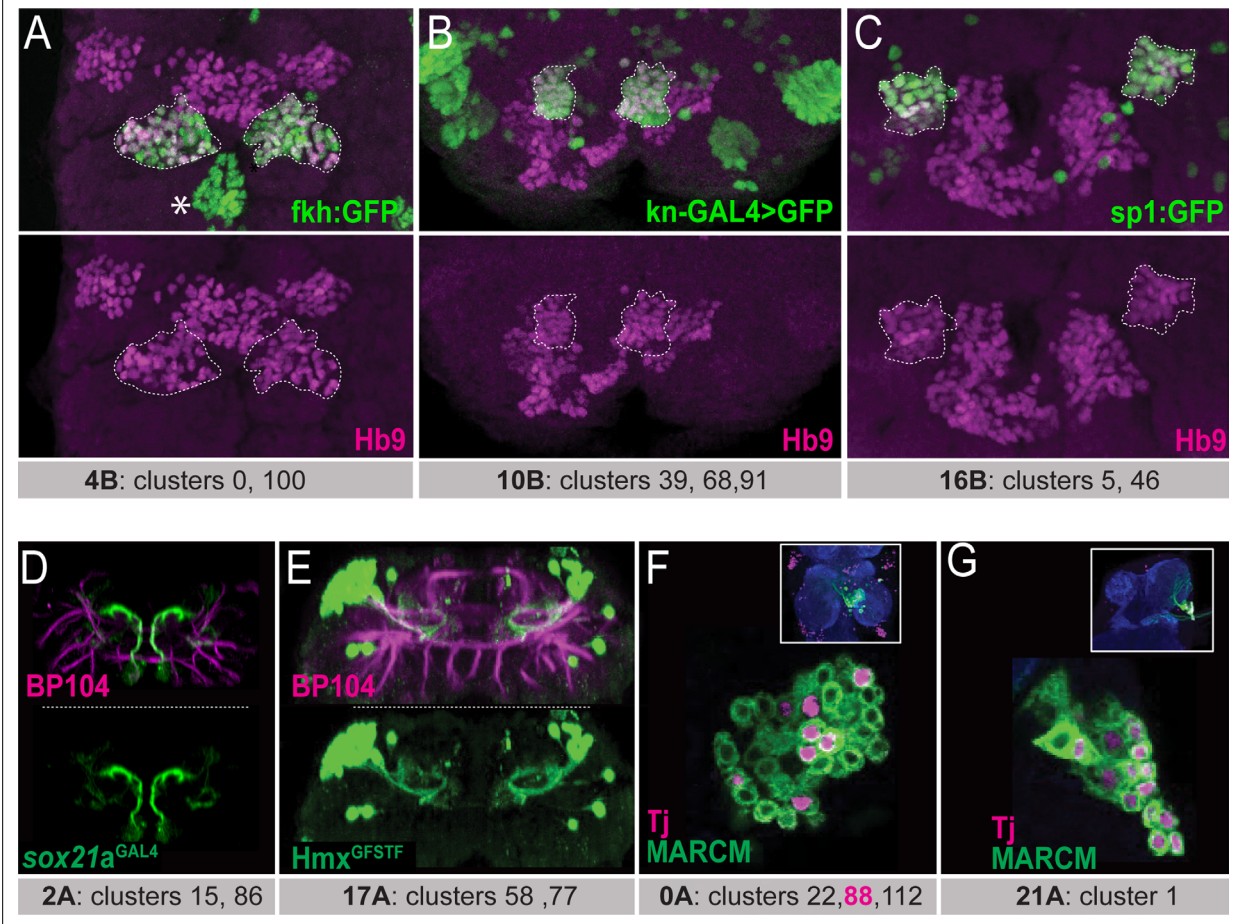

**Figure 2.** Matching the scRNAseq clusters to hemilineages. (**A–C**) Confocal stack of larval VNC displaying the overlapping expressions between transcription factors identified from scRNAseq data (Fkh, Kn, and Sp1; green in (**A**), (**B**), and (**C**), respectively) and Hb9 (magenta) in three lineages: 4B, 10B, and 16B (dashed lines). Asterisk in A indicates the Fkh+Hb9- 0 A lineage neurons. (**D**) Sox21a-GAL4 driven UAS-GFP (green) marks lineage 2 A neurons. (**E**) Hmx^GFSTF reporter (green) marks lineage 17 A neurons. (**F, G**) Wild-type MARCM clones (green) immunostained for Tj (magenta). The insets show the clone location in the VNC counterstained with CadN (blue). (**F**) Tj marks subpopulations of neurons in lineage 0 A in the T2 segment. These neurons likely belong to cluster 88, the only Tj+ 0A cluster in scRNAseq data. (**G**) Tj marks nearly all neurons of lineage 21 A in the T1 segment. Lineage identification of MARCM clones was performed based on neuronal projections detailed in *Truman et al., 2004*; *Kanca et al., 2019*. scRNAseq clusters with the corresponding lineages shown under each panel. Only one thoracic segment is shown. Neuroglian-specific antibody BP104 labels axon bundles of all lineages (magenta in **D**-**E**).

when a cluster marker, or marker combination, is uniquely associated with a hemilineage, it accurately marks this hemilineage.

To identify the clusters that correspond to the remaining 16 hemilineages not assigned by Allen et al., we focused on the orphan clusters, which have not been assigned to any hemilineage. For each of these orphan clusters, we visualized the expression pattern of the cluster marker with reporter genes or antibody staining and studied the morphology of the neurons that expressed this cluster marker. To identify which hemilineage these neurons belonged to, we compared the observed morphology to the documented morphologies of unannotated hemilineages that used the same neurotransmitter as was expressed in the scRNAseq cluster. For example, glutamatergic clusters 15 and 86, which are adjacent in the t-SNE plot, are the only glutamatergic clusters that express Sox21a. To map these clusters to a hemilineage, we studied the morphology of the Sox21a-positive neurons in the VNC by expressing membrane-bound GFP under the control of a CRIMIC line reporting Sox21a expression (*Figure 2D*). This marked a group of ventral and anterior Sox21a-positive neuronal cell bodies situated near the midline in each hemisegment of the larval and adult VNC (*Figure 2D*). Their processes project dorsally and then sharply turn upon reaching the dorsal surface of the neuropil. Based on their glutamatergic neurotransmitter identity and their unique morphology, which matches that of 2 A

interneurons (*Harris et al., 2015*; *Shepherd et al., 2019*), we assigned these clusters to hemilineage 2 A.

Another example is cluster 58, which, among all the VNC hemilineages, uniquely co-expresses *unc-4* and *islet* (also known as tup) (*Lacin et al., 2014*). We had previously studied Unc-4-positive hemilineages and had identified that hemilineage 17 A is the only Unc-4-positive lineage that expresses Islet (*Lacin et al., 2020*). To verify that cluster 58 identified hemilineage 17 A neurons, we examined the expression pattern of another transcription factor, Hmx, which is a cluster marker for cluster 58 (*Allen et al., 2020*). Visualization of Hmx-positive neurons with a CRIMIC line reporting *Hmx* expression revealed that their cell bodies are located on the dorsal surface of the VNC and their processes project into the ipsilateral ventromedial neuropil and then loop dorsally (*Figure 2E*). This morphology is typical of 17 A neurons. Additionally, we found that cluster 77 is marked with the combination of *Hmx* and *tup* and is directly adjacent to cluster 58 in the adult VNC t-SNE plot (*Allen et al., 2020*). Thus, neurons of *Hmx*-positive clusters 58 and 77 likely belong to lineage 17 A (*Figure 2E*). Furthermore, we noted that some transcription factors are expressed in a subset of neurons within a hemilineage and appeared to correspond to one of the multiple scRNAseq clusters assigned to a hemilineage. For example, hemilineage 0 A contains clusters 22, 88, and 112. Of these three, Tj expression is only significant in cluster 88. We generated wild-type MARCM clones of lineage 0 A, and one can see that Tj is expressed in a subset of neurons only, presumably cluster 88 (*Figure 2F*; *Lee and Luo, 1999*). In contrast, other transcription factors (Fkh, Inv, Mab21a, HLH3b, and En) mark all clusters that belong to hemilineage 0 A, as revealed by scRNAseq analysis and our immunostaining-based transcription factor expression analysis (Asterisk in *Figure 2A*; data not shown). In hemilineage 21 A, which is composed of only one scRNAseq cluster, Tj marks nearly all cells (*Figure 2G*). Taken together, these data illustrate how cluster markers identified by scRNAseq data can be used to target individual hemilineages and even distinct subclasses within hemilineages.

Ultimately, we assessed the expression of 23 novel cluster-specific marker genes, all transcription factors, through immunohistochemistry with antibodies against the proteins of interest and/or reporter lines that accurately recapitulate target gene expression (*Table 1*). This effort allowed us to assign at least one cluster to 15 of the 16 previously unassigned hemilineages in the scRNAseq data (*Allen et al., 2020*; *Table 1*). This implies that we now have transcription profiles for 33 of the 34 major hemilineages in the VNC, which facilitates the design of lineage-specific split-GAL4 combinations. The only exception is hemilineage 18B, which remains unassigned to any scRNAseq clusters.

## Building specific and temporally stable driver lines for hemilineages in the VNC

To create a split-GAL4 library that uniquely marks essentially all major hemilineages, we generated gene-specific split-GAL4 driver lines by editing the genomic locus of the transcription factors identified above (*Figure 3*, *Figure 3—figure supplement 1*, Key Resources Table, *Table 1*). To edit the transcription factor locus, we exchanged the intronic cassette of previously engineered MIMIC or CRIMIC lines with a split-GAL4 coding Trojan exon of 13 genes (See Materials and methods). For 11 genes lacking established MIMIC or CRIMIC lines, we used CRISPR/Cas9 mediated gene editing via homology directed repair (HDR) to insert a Trojan exon carrying either DBD or AD split-GAL4 into a coding intron of the target gene and introduced attP sites to facilitate future cassette exchange with any other designer exon via phiC31 mediated cassette exchange (*Diao et al., 2015*; *Nagarkar-Jaiswal et al., 2015*; *Li et al., 2023*; *Figure 3—figure supplement 2*). In select cases, we inserted a Trojan exon directly in frame at the 3' end of the gene (*Figure 3—figure supplement 3*). In total, we generated 34 split-GAL4 lines for 24 genes, 19 using the MiMIC method and 15 using CRISPR editing (Key Resources Table). The CRISPR approach failed only for *tup* and *E5*.

## Comprehensive testing of split-GAL4 combinations to target each hemilineage

Based on our analysis of scRNAseq data, we had clear predictions as to which binary combinations of split-GAL4 lines would label which hemilineages. To test these predictions, we specifically paired these new split-GAL4 lines either with one another or with previously generated split-GAL4 lines (*Table 1*; *Lacin et al., 2019*; *Lacin et al., 2024*; *Chen et al., 2023a*). Reconstituted GAL4 was visualized by UAS-myr-GFP or tdTomato and compared to the typical hemilineage morphologies of cell bodies and

**Table 1.** Overview of cluster annotation, lineage-specific marker genes, and tested split-GAL4 driver lines.

| Lineage | Clusters (Allen et al.) | Markers | Driver line combinations |
|---|---|---|---|
| 0A | 22, 88, 112 | En, Inv, Fkh, Tj, Lim1, grn, HLH3B, Mab-21, Gad1 | inv-GAL4-DBD, tj-p65.AD: * * * * fkh-GAL4-DBD, tj-p65.AD: * * * * mab21-p65.AD, fkhGAL4-DBD: * * * |
| 1A | 16 | Dr, Ets21C, Ptx1, ChAT | Dr-p65.AD, ets21C-GAL4-DBD: * * * |
| 1B | 12, 47 | HLH4C, H15, Mid, Gad1 | HLH4C-GAL4-DBD, H15-p65.AD: * * * |
| 2A | 15, 86 | HLH3B, Oc, Sox21a, Drgx, Lim1, grn, svp, VGlut | sox21a-GAL4-DBD, VGlut-p65.AD: * * * * sox21a-GAL4-DBD, lim1-VP16.AD: * * * |
| 3A | 7, 37, 85 | H15, HGTX, Grn, Lim1, ChAT | H15-p65.AD, ChaT-GAL4-DBD: * |
| 3B | 26 | Fer3, CG4328, Gad1 | fer3-GAL4-DBD, cg4328-p65.AD: * |
| 4B | 0, 100 | Exex, Ap, Fkh, Tey, HGTX, HLH4C, Oc, ChAT | ap-p65.AD, fkhGAL4-DBD: * * * ap-p65.AD, hgtx-GAL4-DBD: * * * * |
| 5B | 20, 87, 97 | Vg, Toy, Vsx2, Lim1, Gad1 | vg-p65.AD, toy-GAL4-DBD: * * * * |
| 6A | 9, 28 | Mab-21, Toy, Gad1 | mab21-p65.AD, toy-GAL4-DBD: * * |
| 6B | 3, 89 | Vg, Sens-2, En, CG4328, Vsx2, Gad1 | sens2-p65.AD, vg-GAL4-DBD sens2-GAL4-DBD, vg-p65.AD: * * CG4328-p65.AD, vg-GAL4-DBD: * * * |
| 7B | 2, 62 | Unc-4, Sv, Mab-21, ChAT | unc-4-p65.AD, mab21-GAL4-DBD: * * * unc-4-GAL4-DBD, sv-p65.AD: * * * |
| 8A | 6, 69, 110 | Ey, Ems, Toy, Ets65A, VGluT | ems-GAL4-DBD, eyAD: * * * * ems-GAL4-DBD, toy-p65.AD: * * ems-GAL4-DBD, vGluT-p65.AD: * * * |
| 8B | 8, 53, 76 | C15, Lim3, Acj6, ChAT | C15-p65.AD, lim3-GAL4-DBD: * * * |
| 9A | 31, 50, 56, 57 | Dr, Ets65A, grn, sox21a, Gad1 | Dr-p65.AD, gad1-GAL4-DBD: * * * * Dr-p65.AD, sox21a-GAL4-DBD: * * * * |
| 9B | 54, 76 | Lim3, Drgx, Sens-2, Acj6, Tup, HLH4C, VGluT | acj6-p65.AD, VGluT-GAL4-DBD: * * * |
| 10B | 39, 68, 91 | Exex, Kn, Sens-2, Lim3, ChAT | knot-p65.AD, hb9-GAL4-DBD: * * * * hb9-p65.AD, sens-2-GAL4-DBD: * * * * knot-p65.AD, nkx6-GAL4-DBD: * * * * knot-p65.AD, lim3-GAL4-DBD: * * * |
| 11 A | 21 | Unc-4, Tey, ChAT | unc-4-GAL4-DBD, tey-VP16: * * * unc-4-p65.AD, hgtx-GAL4-DBD: * * * |
| 11B | 38 | Eve, HLH4C, Gad1 | eve-p65.AD, gad1-GAL4-DBD: * * * * |
| 12 A | 40 | Unc-4, TfAP-2, Grn, ChAT | unc-4-GAL4-DBD, TfAP2-p65.AD: * * * |
| 12B | 30, 73, 81, 83, 94 | Fer3, HGTX, CG4328, H15, Tey, Gad1 | HGTX-GAL4-DBD, gad1-p65.AD: * * |
| 13 A | 48, 75, 79 | Dbx, Fer2, Dmrt99B, Gad1 | dbx-GAL4-DBD, dmrt99B-p65.AD: * * |
| 13B | 17, 25 | D, Vg, CG4328, tey, svp, Gad1 | vg-GAL4-DBD, D-VP16.AD: * * vg-GAL4-DBD, tey-VP16.AD: * * * |
| 14 A | 13, 41, 74 | Dr, Toy, Lim1, Ets65A, Grn, VGluT, | Dr-p65.AD, toy-GAL4-DBD: * * * |
| 15B | 36, 52, 80 | Tup, Lim3, HGTX, VGlut | HGTX-GAL4-DBD, VGlut-p65.AD: * * * nkx6- GAL4-DBD, twit-p65.AD: * * * |
| 16B | 5, 46 | Lim3, Exex, Bi, Sp1, VGlut, | hb9-p65.AD, bi-GAL4-DBD: * * * hb9-p65.AD, VGlut-GAL4-DBD: * * * |
| 17 A | 58, 77 | Unc-4, Hmx, Tup, ChAT | unc-p65.AD, hmx-GAL4-DBD: * * * * |
| 18B | N/A | Unc-4, ChAT | **No line** |
| 19 A | 19, 59, 82 | Dbx, Fer2, Scro, Gad1 | dbx-GAL4-DBD, scro-p65.AD: * * * |
| 19B | 27, 71 | Unc-4, Otp, ChAT | **No line** |
| 20/22 A | 14, 33, 34, 78, 108 | Bi, Ets65A, Sv, ChAT | sv-p65.AD, ets65-GAL4-DBD: * * * bi-GAL4-DBD, shaven-p65.AD: * * bi-p65.AD, ets65A-GAL4-DBD: * * |
| 21 A | 1 | Dr, Ey, Tj, VGluT | Dr-p65.AD, tj-GAL4-DBD: * * * * Dr-p65.AD, ey-GAL4-DBD: * * * |
| 23B | 35, 51, 67, 93 | Unc-4, Acj6, Slou, Otp, ChAT | unc-4-p65.AD, acj6-GAL4-DBD: * * * |
| 24B | A small subset of clusters 52 and 36 | Toy, Ems, Twit, VGlut | ems-GAL4-DBD, twit-p65.AD: * * * |

*Table 1 continued on next page*

*Table 1 continued*

| Lineage | Clusters (Allen et al.) | Markers | Driver line combinations |
|---------|-------------------------|---------|--------------------------|

**** Very specific for one hemilineage; *** Specific, some contamination from other neurons; ** Somewhat specific, significant contribution of e.g. motor neurons or sensory neurons; * More than one hemilineage marked

axonal trajectories to assess whether the split-GAL4 line targeted the predicted lineage. We identified 44 split-GAL4 combinations that target 32 out of 34 hemilineages and summarize the expression pattern of each combination in *Table 1* and *Supplementary file 1*. *Figure 3*, *Figure 3—figure supplement 1* display the larval and adult VNC expression patterns of the driver lines generated for 32 out of 34 hemilineages. Robust expression was also observed in 27 hemilineages in the larva, making these lines suitable for tracking the developmental history of their respective hemilineage during metamorphosis. The expression patterns of the split-GAL4 combinations for the remaining lineages (1B,3B,13A,13B, and 24B) start during pupal stages. (1B: *HLH4C*-GAL4$^{DBD}$, *H15*-GAL4$^{AD}$; 3B: *H15*-GAL4$^{AD}$, *ChAT*-GAL4$^{DBD}$; 13 A: *dbx*-GAL4$^{DBD}$, *dmrt99B*-GAL4$^{AD}$; 13B: *vg*-GAL4$^{DBD}$, *d*-GAL4$^{AD}$ or *vg*-GAL4$^{DBD}$, *tey*-GAL4$^{AD}$; 24B: *ems*-GAL4$^{DBD}$, *twit*-GAL4$^{AD}$, data not shown).

## Application of developmentally stable hemilineage specific split-GAL4 lines

### Morphological changes of 4B neurons during development

To show the applicability of our driver lines for developmental studies, we characterized the morphological changes in the neuronal processes during metamorphosis. Neurons of hemilineage 4B are excitatory cholinergic local interneurons (*Lacin et al., 2019*; *Shepherd et al., 2019*), and their arborizations are restricted to the ipsilateral leg neuropils and directly synapse with leg motor neurons in addition to many interneurons (*Marin et al., 2024*; *Lesser et al., 2024*). We have built three different combinations of split-GAL4 lines, each of which specifically targets most, if not all, post-embryonic 4B neurons. Two of these drivers, *ap*-GAL4$^{AD}$ with *fkh*-GAL4$^{DBD}$ and *ap*-GAL4$^{AD}$ with *HLC4C*-GAL4-$^{DBD}$, drive reporter expression in 4B neurons starting from early larval stages while *ap*-GAL4$^{AD}$ with *HGTX*-GAL4$^{DBD}$ drives robust expression beginning at the white pupal stage (data not shown). Due to its stronger and earlier expression pattern, we used *ap*-GAL4$^{AD}$ with *fkh*-GAL4$^{DBD}$ to mark the morphology of 4B neurons at 0, 3, 12, 24, 48, and 72 hr APF (*Figure 4*). Like all the other post-embryonic neurons, 4B neurons extend an initial simple neurite bundle after they are born and do not show any further arborization until metamorphosis. As seen in the VNC of a 0 hr APF animal, this initial 4B neurite bundle projects dorsally away from the cell bodies and innervates the leg neuropil diagonally across the dorso-ventral axis (*Figure 4A and A'*). At 3 hr APF, multiple growth cones that point in different directions are visible on the tip of the 4B bundle (*Figure 4B and B'*). At 12 hr APF, these growth cones transform into three distinct branches, extending medially, laterally, or dorsally (*Figure 4C and C'*). At 24 hr APF, finer processes extend from these branches and puncta-like staining is visible, which suggests that synapses are being formed (*Figure 4D and D'*). At 48 hr APF, the crowded, finer processes are resolved into more refined and discreet processes and the synaptic puncta-like staining becomes more extensive (*Figure 4E and E'*). At 72 hr APF, the synaptic puncta appear to increase in size and to take on bouton-like shapes, resembling the adult morphology (*Figure 4F and F'*). In summary, by employing a developmentally stable driver line specific to hemilineage 4B, we documented stepwise morphological changes in the outgrowth of neuronal processes of 4B neurons throughout metamorphosis. These changes occur in distinct phases: initial neurite bundle expansion through new branch additions, followed by the formation and refinement of finer processes and synapses, and concluding with synaptic growth.

### Neurotransmitter use

Another use for our library of transcription factor-specific Split-GAL4 lines is to determine which neurotransmitter a specific neuronal population produces. For example, we identified which neurotransmitter Acj6-positive neurons produce. We combined *Acj6*-split-GAL4 with a split-GAL4 line reporting the expression of one of the neurotransmitter marker genes -Gad1, ChAT, or VGlut- to

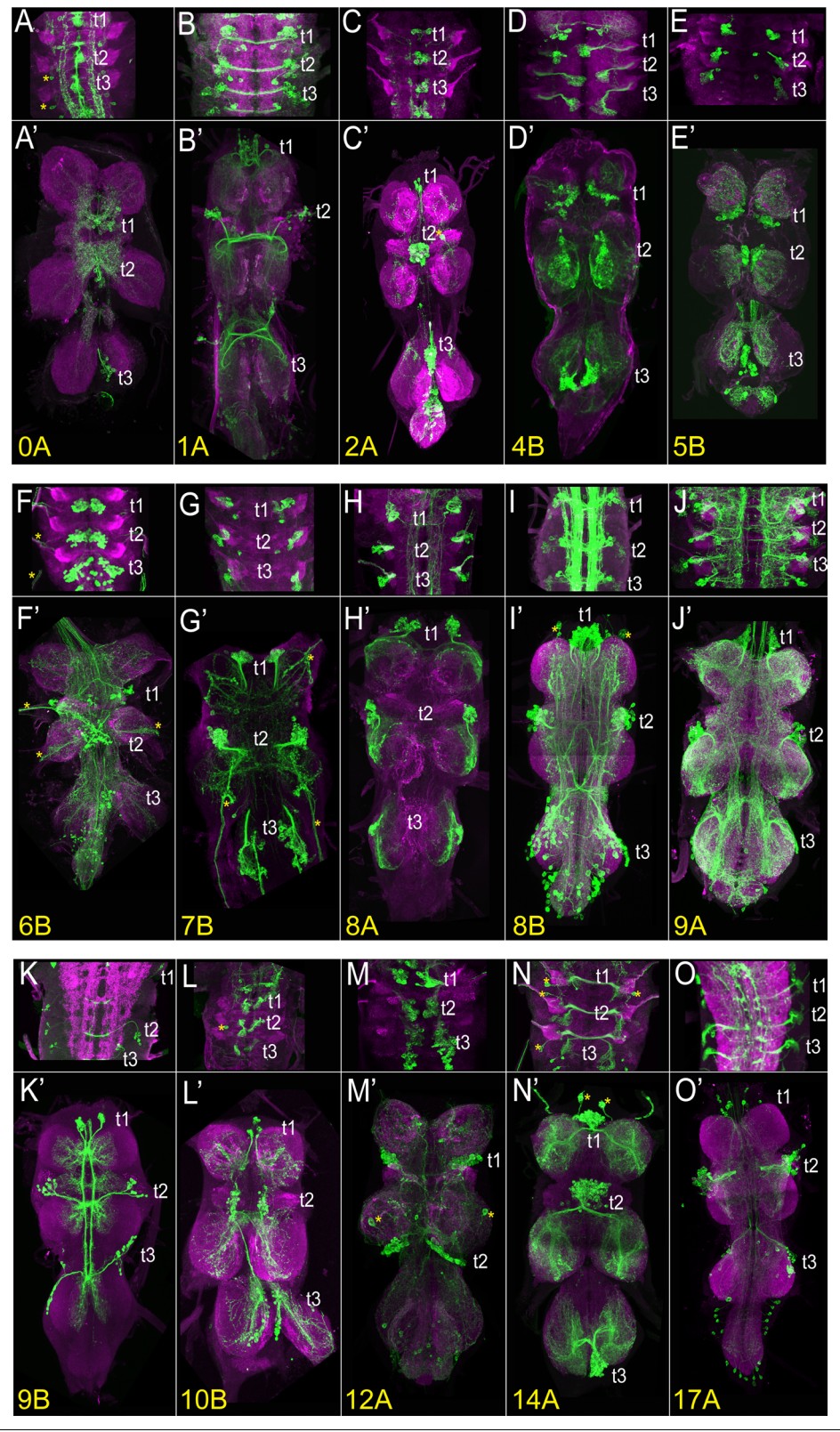

**Figure 3.** The VNC expression of select driver lines from the split-GAL4 library targeting individual hemilineages. Projections of confocal stacks showing the expression pattern of split-GAL4-driven membranous GFP (green) in the larval (**A–O**) and adult VNC (**A′-O′**). Only thoracic segments are shown in the larval images. (**A, A′**) Hemilineage 0 A, marked by *inv*-GAL4-DBD, *tj*-VP16.AD. (**B, B′**) Hemilineage 1 A marked by *ets21c*-GAL4-DBD, *Dr*-p65.AD.

*Figure 3 continued on next page*

*Figure 3 continued*

(**C, C'**) Hemilineage 2 A marked by sox21a GAL4-DBD, *VGlut*-p65.AD. (**D, D'**) Hemilineage 4B marked by *ap*-p65.AD, *fkh*-GAL4-DBD. (**E, E'**) Hemilineage 5B marked by *vg*-p65.AD, *toy*-GAL4-DBD. (**F, F'**) Hemilineage 6B marked by *sens2*-p65.AD, *vg*-GAL4-DBD. (**G, G'**) Hemilineage 7B marked by *mab21*-GAL4-DBD, *unc-4*-p65.AD. (**H**) Hemilineage 8 A marked by *ems*-GAL4-DBD, *ey*-p65.AD. (**I, I'**) Hemilineage 8B marked by *lim3*-GAL4-DBD, *C15*-p65.AD. (**J, J'**) Hemilineage 9 A marked by *Dr*-p65.AD, *gad1*-GAL4-DBD. (**K, K'**) Hemilineage 9B marked by *acj6*-p65.AD, *VGlut*-GAL4-DBD. (**L, L'**) Hemilineage 10B marked by *knot*-p65.AD, *hb9*-GAL4-DBD. (**M, M'**) Hemilineage 12 A marked by *TfAP-2*-GAL4-DBD, *unc-4*-p65.AD. (**N, N'**) Hemilineage 14 A marked by *Dr*-p65.AD, *toy*-GAL4-DBD. (**O, O'**) Hemilineage 17 A marked by *unc-4*-p.65AD, *hmx*-GAL4-DBD. The VNC was counterstained with CadN (magenta). The target lineage is indicated on the left bottom corner of each panel. Z-projections were made of selected regions of the VNC to highlight the cell-body clustering and axonal budling.

The online version of this article includes the following video and figure supplement(s) for figure 3:

**Figure supplement 1.** The rest of the driver lines from the Split-GAL4 library targeting individual hemilineages.

**Figure supplement 2.** CRISPR-mediated insertion of Trojan Exons.

**Figure supplement 3.** Direct tagging with CRISPR.

**Figure 3—video 1.** Hemilineage 1 A activation on a decapitated animal, 60 FPS.
https://elifesciences.org/articles/106042/figures#fig3video1

**Figure 3—video 2.** Hemilineage 1 A activation on an intact animal, 50 FPS.
https://elifesciences.org/articles/106042/figures#fig3video2

**Figure 3—video 3.** Hemilineage 1B activation on a decapitated animal, 40 FPS.
https://elifesciences.org/articles/106042/figures#fig3video3

**Figure 3—video 4.** Hemilineage 1B activation on an intact animal, 40FPS.
https://elifesciences.org/articles/106042/figures#fig3video4

**Figure 3—video 5.** Hemilineage 2 A activation on a decapitated animal, 60FPS.
https://elifesciences.org/articles/106042/figures#fig3video5

**Figure 3—video 6.** Hemilineage 2 A activation on an intact animal, 40 FPS.
https://elifesciences.org/articles/106042/figures#fig3video6

**Figure 3—video 7.** Hemilineage 4B activation on a decapitated animal, 72FPS.
https://elifesciences.org/articles/106042/figures#fig3video7

**Figure 3—video 8.** Hemilineage 4B activation on an intact animal, 72FPS.
https://elifesciences.org/articles/106042/figures#fig3video8

**Figure 3—video 9.** Hemilineage 5B activation on a decapitated animal, 60FPS.
https://elifesciences.org/articles/106042/figures#fig3video9

**Figure 3—video 10.** Hemilineage 5B activation on an intact animal, 50 FPS.
https://elifesciences.org/articles/106042/figures#fig3video10

**Figure 3—video 11.** Hemilineage 5B activation on an intact feeding animal, 25FPS.
https://elifesciences.org/articles/106042/figures#fig3video11

**Figure 3—video 12.** Hemilineage 5B activation on an intact animal-tethered flight, 25FPS.
https://elifesciences.org/articles/106042/figures#fig3video12

**Figure 3—video 13.** Hemilineage 5B activation on an intact animal walking, 25FPS.
https://elifesciences.org/articles/106042/figures#fig3video13

**Figure 3—video 14.** Hemilineage 6B activation on a decapitated animal 40 FPSS.
https://elifesciences.org/articles/106042/figures#fig3video14

**Figure 3—video 15.** Hemilineage 6B activation on an intact animal-tethered flight, 81FPS.
https://elifesciences.org/articles/106042/figures#fig3video15

**Figure 3—video 16.** Hemilineage 7B activation on a decapitated animal, 40FPS.
https://elifesciences.org/articles/106042/figures#fig3video16

**Figure 3—video 17.** Hemilineage 7B activation on a decapitated animal, 500FPS-5Xslower.
https://elifesciences.org/articles/106042/figures#fig3video17

**Figure 3—video 18.** Hemilineage 7B activation on an intact animal, 40FPS.
https://elifesciences.org/articles/106042/figures#fig3video18

*Figure 3 continued*

**Figure 3—video 19.** Hemilineage 8 A activation on a decapitated animal, 40FPS.
https://elifesciences.org/articles/106042/figures#fig3video19

**Figure 3—video 20.** Hemilineage 8 A activation on an intact animal, 40FPS.
https://elifesciences.org/articles/106042/figures#fig3video20

**Figure 3—video 21.** Hemilineage 8B activation on a decapitated animal, 500FPS-10Xslower.
https://elifesciences.org/articles/106042/figures#fig3video21

**Figure 3—video 22.** Hemilineage 8B activation on an intact animal, 500FPS-10Xslower.
https://elifesciences.org/articles/106042/figures#fig3video22

**Figure 3—video 23.** Hemilineage 9 A activation on a tethered decapitated animal, 40FPS.
https://elifesciences.org/articles/106042/figures#fig3video23

**Figure 3—video 24.** Hemilineage 9 A activation on a decapitated animal, 40FPS.
https://elifesciences.org/articles/106042/figures#fig3video24

**Figure 3—video 25.** Hemilineage 9 A activation on an intact animal, 40FPS.
https://elifesciences.org/articles/106042/figures#fig3video25

**Figure 3—video 26.** Hemilineage 9B activation on a decapitated animal, 40FPS.
https://elifesciences.org/articles/106042/figures#fig3video26

**Figure 3—video 27.** Hemilineage 9B activation on an intact animal, 40FPS.
https://elifesciences.org/articles/106042/figures#fig3video27

**Figure 3—video 28.** Hemilineage 10B activation on a decapitated animal, 60FPS.
https://elifesciences.org/articles/106042/figures#fig3video28

**Figure 3—video 29.** Hemilineage 10B activation on an intact animal, 50FPS.
https://elifesciences.org/articles/106042/figures#fig3video29

**Figure 3—video 30.** Hemilineage 11 A activation with a strong stimulation on a decapitated animal, 500FPS-10Xslower.
https://elifesciences.org/articles/106042/figures#fig3video30

**Figure 3—video 31.** Hemilineage 11 A activation with a weak stimulation on a decapitated animal, 500FPS.
https://elifesciences.org/articles/106042/figures#fig3video31

**Figure 3—video 32.** Hemilineage 11 A activation with a strong stimulation on an intact animal, 40FPS.
https://elifesciences.org/articles/106042/figures#fig3video32

**Figure 3—video 33.** Hemilineage 11 A activation with a weak stimulation on an intact animal, 40FPS.
https://elifesciences.org/articles/106042/figures#fig3video33

**Figure 3—video 34.** Hemilineage 11B activation on a decapitated animal, 40FPS.
https://elifesciences.org/articles/106042/figures#fig3video34

**Figure 3—video 35.** Hemilineage 11B activation on an intact animal, 40FPS.
https://elifesciences.org/articles/106042/figures#fig3video35

**Figure 3—video 36.** T1 clonal activation of hemilineage 12 A neurons on a decapitated animal, sample 1, 100FPS.
https://elifesciences.org/articles/106042/figures#fig3video36

**Figure 3—video 37.** T1 clonal activation of hemilineage 12 A neurons on a decapitated animal, sample 2, 100FPS.
https://elifesciences.org/articles/106042/figures#fig3video37

**Figure 3—video 38.** Hemilineage 13 A activation on a decapitated animal, 40FPS.
https://elifesciences.org/articles/106042/figures#fig3video38

**Figure 3—video 39.** Hemilineage 13 A activation on two intact animals, 40FPS.
https://elifesciences.org/articles/106042/figures#fig3video39

**Figure 3—video 40.** Hemilineage 13B activation on a decapitated animal, 40FPS.
https://elifesciences.org/articles/106042/figures#fig3video40

**Figure 3—video 41.** hemilineage 13B activation on an intact animal, 40FPS.
https://elifesciences.org/articles/106042/figures#fig3video41

**Figure 3—video 42.** Hemilineage 14 A activation on a decapitated animal, 60FPS.
https://elifesciences.org/articles/106042/figures#fig3video42

**Figure 3—video 43.** Hemilineage 14 A activation on an intact animal, 40FPS.

*Figure 3 continued*

https://elifesciences.org/articles/106042/figures#fig3video43

**Figure 3—video 44.** hemilineage 15B activation on a decapitated animal, 50FPS.

https://elifesciences.org/articles/106042/figures#fig3video44

**Figure 3—video 45.** hemilineage 15B activation on an intact animal, 40FPS.

https://elifesciences.org/articles/106042/figures#fig3video45

**Figure 3—video 46.** hemilineage 16B activation on a decapitated animal, 60FPS.

https://elifesciences.org/articles/106042/figures#fig3video46

**Figure 3—video 47.** hemilineage 16B activation on an intact animal, 40FPS.

https://elifesciences.org/articles/106042/figures#fig3video47

**Figure 3—video 48.** hemilineage 17 A activation on a decapitated animal, 40FPS.

https://elifesciences.org/articles/106042/figures#fig3video48

**Figure 3—video 49.** Hemilineage 17 A activation on an intact animal, 40FPS.

https://elifesciences.org/articles/106042/figures#fig3video49

**Figure 3—video 50.** Hemilineage 19 A activation on a decapitated animal, 40FPS.

https://elifesciences.org/articles/106042/figures#fig3video50

**Figure 3—video 51.** Hemilineage 19 A activation on an intact animal, 40FPS.

https://elifesciences.org/articles/106042/figures#fig3video51

**Figure 3—video 52.** Hemilineage 21 A activation on a decapitated animal, 25FPS.

https://elifesciences.org/articles/106042/figures#fig3video52

**Figure 3—video 53.** Hemilineage 21 A activation on an intact tethered animal, 200FPS.

https://elifesciences.org/articles/106042/figures#fig3video53

**Figure 3—video 54.** Hemilineage 23B activation on a decapitated animal, 33FPS.

https://elifesciences.org/articles/106042/figures#fig3video54

**Figure 3—video 55.** Hemilineage 23B activation on an intact animal, 47FPS.

https://elifesciences.org/articles/106042/figures#fig3video55

visualize GABAergic, cholinergic, and glutamatergic Acj6-positive neurons, respectively (*Figure 5*). Acj6 is known to be expressed in glutamatergic 9B and cholinergic 8B and 23B hemilineages (*Lacin et al., 2019*). As expected, in each thoracic hemisegment of the VNC, these split-GAL4 combinations marked a single cluster of glutamatergic Acj6-positive neurons corresponding to 9B neurons (arrowheads in *Figure 5A*), two clusters of cholinergic Acj6-positive neurons corresponding to 8B and 23B

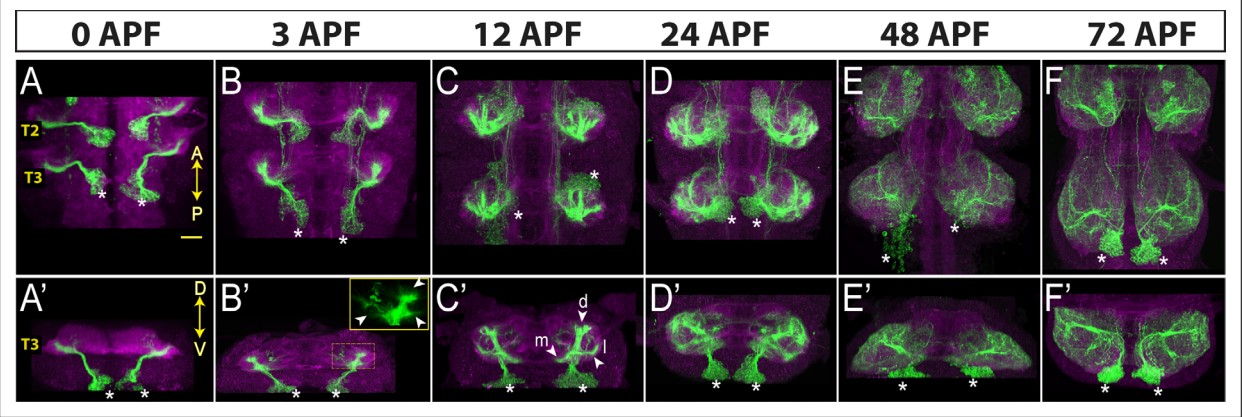

**Figure 4.** Neurons of hemilineage 4B show profound morphological changes during development. Projection of confocal stacks showing the morphology of 4B neurons (green) marked with the *ap*-GAL4^AD and *fkh*-GAL4^DBD driver combination across different developmental time points during metamorphosis: 0, 3, 12, 24, and 48 hr after puparium formation (APF). The VNC is counterstained with CadN (magenta). Cell bodies of 4B neurons in the T3 region are marked with asterisks. (**A–F**) Complete projections in T2-T3 segments. Anterior (**A**) up; posterior (**P**) down. (**A'-F'**) Transverse views of the entire T3 segments across the dorso-ventral (**D–V**) axis; Dorsal is up. Arrowheads in B' mark growth cones. Arrowheads in C' mark three new branches towards the medial (**m**), lateral (**l**) and dorsal (**d**) part of the leg neuropil. Scale bar is 20 micron.

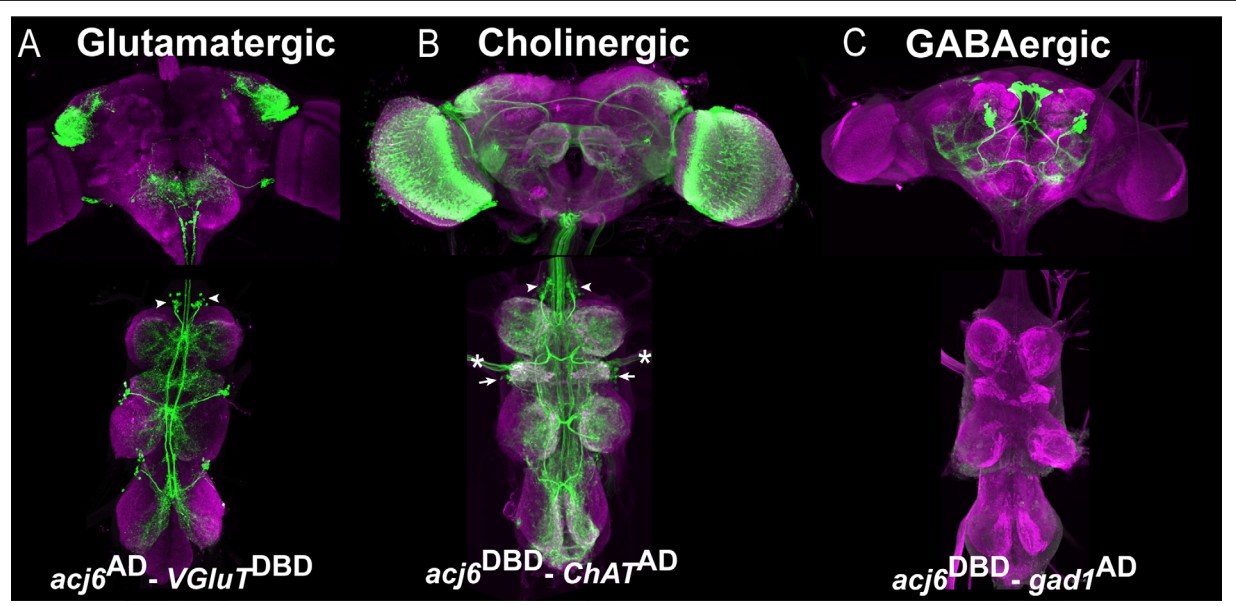

**Figure 5.** Acj6-positive neurons in the VNC are glutamatergic or cholinergic. (**A–C**) Split-GAL4 line reporting Acj6 expression intersected with a cognate split-GAL4 line reporting the expression of Gad1, ChAT or VGlut to visualize GABAergic, cholinergic, and glutamatergic populations of Acj6-positive neurons, respectively. The VNC is counterstained with CadN (magenta). (**A**) Split-GAL4 combination acj6-GAL4^AD, gad1-GAL4^DBD>UAS-GFP driven UAS-GFP shows that the optic lobes contain cholinergic Acj6-positive neurons in addition to a few clusters of neurons with prominent long projections. In the VNC, two cholinergic clusters per hemisegment corresponding to 8B (arrowheads) and 23B (arrows) hemilineages are labeled in addition to some sensory neurons (asterisks). (**B**) Split-GAL4 combination acj6-GAL4^AD, VGlut-GAL4^DBD> UAS-GFP marks a single glutamatergic lineage in the dorsal part of the brain and one 9 A glutamatergic cluster in the VNC. (**C**) Split-GAL4 combination acj6-GAL4^AD, gad1-GAL4^DBD>UAS-GFP marks two GABAergic lineages in the brain and nothing in the VNC.

neurons (arrowhead and arrows, respectively in *Figure 5B*), and no GABAergic Acj6-positive neurons (*Figure 5C*). In each hemibrain, the same split-GAL4 combinations identified one glutamatergic cluster with local projections, several cholinergic clusters with long projections, and two GABAergic clusters with long projections. Additionally, we detected cholinergic Acj6-positive leg and antennal sensory neurons and optic lobe neurons (*Figure 5A–C*).

Furthermore, one can quickly test if the transcription factor that drives split-GAL4 expression is required for neurotransmitter production. When one inserts Trojan split-GAL4 hemidrivers in the first intron of the transcription factor gene locus, the Hsp70 transcriptional terminator located at the 5' end of the trojan exon prematurely ends the transcript, and the truncated transcription factor oftentimes acts as a null mutant. We leverage this to test whether Acj6 has any role in the neurotransmitter identity of these neurons by using *acj6* split lines. We repeated the same experimental procedure in an *acj6* mutant background and found no apparent differences in neurotransmitter expression, concluding that Acj6 is dispensable for neurotransmitter identity (not shown).

In conclusion, we show that one can quickly assay neuronal identity features such as neurotransmitters and their receptors in specific populations of neurons in the entire CNS by simply using the split-GAL4 system and intersecting the expression patterns of a lineage-specific gene with the expression of another gene coding for neuronal identity. However, care should be taken interpreting these results, as the presence of the Hsp70 transcriptional terminator at the 5' end of the Trojan exon may also affect the host gene's 3' UTR regulation, such that these drivers do not always faithfully recapitulate expression of genes subject to post-transcriptional regulation as was observed for ChAT (*Lacin et al., 2019*; *Chen et al., 2023b*).

## Behavioral analysis with targeted lineage manipulation

Understanding the functional roles of specific hemilineages in the VNC is crucial for unraveling the neural circuits that govern behavior, yet the tools to study these lineages in detail have been limited. *Harris et al., 2015* developed genetic tools to mark and track a small subset of VNC hemilineages

through metamorphosis into the adult. This work used thermogenetic methods to stimulate neuronal activity of specific hemilineages to assess their function. This was done with decapitated flies to remove the effect of driver line expression in the brain. However, for many hemilineages, either no driver line existed or only a small portion of a hemilineage was targeted. To overcome these issues, we use our new split-GAL4 combinations to manipulate eight hemilineages for which no drivers previously existed (0 A, 1B, 4B,8B, 9B, 14 A, 16B, 17 A) and to target 16 lineages studied by *Harris et al., 2015* with better coverage (*Table 2*). We evaluate lineage-coupled behavior with optogenetic activation, a method that is more robust and has a better time resolution compared to thermogenetic activation (*Klapoetke et al., 2014*). We show below how this approach is compatible with genetic methods to remove unwanted GAL4-mediated gene expression in the brain by applying a teashirt/FLP-based genetic intersection with the LexA/LexAop system to restrict GAL4 expression to the VNC (*Simpson, 2016*). A major advantage of such a layered genetic set-up is that behavior can be evaluated in intact flies without the need for decapitation. The lineage-behavior analysis of the 26 hemilineages is summarized in *Table 2* and the response of intact and decapitated flies upon optogenetic activation is shown in *Figure 3—videos 1–55*. In the sections below, we summarize four examples.

## Hemilineage 8B

Hemilineage 8B neurons, which are cholinergic and excitatory, show complex segment-specific intersegmental projections that innervate the tectulum and leg neuropil (*Shepherd et al., 2019*). To target 8B neurons, we used *lim3*-GAL4$^{DBD}$ and *c15*-GAL4$^{AD}$, which target most of the 8B neurons as well as numerous neuronal clusters in the brain (*Figure 6A*). We activated only 8B neurons through exclusion of brain neurons by layering *lim3*-GAL4$^{DBD}$, *c15*-GAL4$^{AD}$ with a *teashirt (tsh)* driver that restricts expression of the optogenetic construct CsChrimson-mVenus to only VNC neurons (*Simpson, 2016*; *Figure 6B*). We observed that optogenetic stimulation of 8B neurons triggered jump behavior in intact and decapitated animals (*Figure 6C and D*, *Figure 3—video 19*, *Figure 3—video 20*). Unlike 7B neuronal activation, which makes flies raise their wings before jumping (*Figure 3—video 16*, *Figure 3—video 17*, *Figure 3—video 18*; *Lacin et al., 2020*), 8B activation resulted in jumping without a wing raise, which is reminiscent of the Giant Fiber (GF) induced escape movement sequence (*Namiki et al., 2018*; *Namiki et al., 2022*; *Zabala et al., 2009*; *Card and Dickinson, 2008*; *Cheong et al., 2023*). Therefore, our results suggest that 8B neurons participate in the GF-driven take-off circuit.

To investigate the relationship between 8B and the GF neurons, we analyzed the synaptic connections of the GF (DPN01) using MANC2.1 in neuPrint (*Marin et al., 2024*; *Plaza et al., 2022*; *Takemura et al., 2023*), and focused on neurons with at least five synapses, for one half of the bilateral symmetric circuit. We found that hemilineage 8B neurons are upstream synaptic partners of the GF, with 12 8B neurons accounting for 12.5% of the GF synaptic inputs (*Figure 6—figure supplement 1*, *Supplementary file 2*). Surprisingly, 8B neurons were also downstream synaptic partners of the GF, with 13 neurons accounting for 12.5% of the GF's synaptic outputs (*Figure 6—figure supplement 1*, *Supplementary file 3*). This contribution is significant, as it is even higher than the 8.7% of synaptic output connections that a GF dedicates to innervating the tergotrochanter motor neuron (TTMn), which innervates the jump muscle. We next compared if those 8B neurons that are downstream partners of the GF also provide input to the GF. Surprisingly, the majority of 8B neurons that connect to the GF are both downstream and upstream synaptic partners. These nine neurons make up 21.5% and 9.1% of total GF synaptic inputs and outputs, respectively. Taken together, our behavioral data and the connectome analysis suggest that a subset of 8B neurons functions in the GF circuit and elicits take-off behavior.

## Hemilineage 9A

Hemilineage 9 A is composed of inhibitory GABAergic neurons, which integrate sensory input from leg proprioceptive neurons (*Agrawal et al., 2020*; *Lacin et al., 2019*). To activate 9 A neurons, we drove CsChrimson expression with *Dr*-GAL4$^{AD}$ and *gad1*-GAL4$^{DBD}$. Decapitated animals exhibited erratic walking behavior with their legs extended when the stimulus lasted over three seconds, and this erratic walking immediately stopped when the stimulus ended (*Figure 6E*, *Figure 3—video 23*, *Figure 3—video 24*, *Figure 3—video 25*). In agreement with previous reports (*Agrawal et al., 2020*;

**Table 2.** Overview of behavioral phenotypes upon optogenetic activation of specific hemilineages.

| Lineage | *Genotype*: Phenotype | Videos |
|---|---|---|
| 0A | *tj-p65.AD, inv-GAL4-DBD*: No apparent behavioral response observed in response to acute optogenetic activation. | N/A |
| 1A | *Dr-p65.AD, Ets21C-GAL4-DBD*: Activation in both intact and decapitated animals drove leg extension making fly taller. Our observation differed from previously observed phenotypes of erratic forward locomotion, occasionally interrupted by grooming in decapitated animals (*Harris et al., 2015*). | *Figure 3—video 1*; *Figure 3—video 2* |
| 1B | *H15-p65.AD, HLH4C-GAL4-DBD*: Activation in both intact and decapitated flies drives leg rotational movement causing the joint between the femur and tibia to bend laterally, most pronounced by the hind legs. | *Figure 3—video 3*; *Figure 3—video 4* |
| 2A | *VGlut-p65.AD, Sox21a-GAL4-DBD*: Activation in intact animals drove high-frequency wing flapping, consistent with the findings of Harris et al which showed the same phenotype with the decapitated flies. In our experiments with decapitated animals, no wing buzzing was observed, and only halteres moved ventrally upon stimulation. | *Figure 3—video 5* ; *Figure 3—video 6* |
| 4B | *ap-p65.AD, HGTX-GAL4-DBD*: Activation causes a full extension of all the legs in both decapitated and intact flies. | *Figure 3—video 7*; *Figure 3—video 8* |
| 5B | *vg-p65.AD, toy-GAL4-DBD*: Activation of 5B neurons halts almost every movement in the animal, causing walking, grooming, flying (tethered flight assay), and feeding flies to halt these behaviors. Decapitated animals also halt their grooming activity in response to 5B activation. Active 5B neurons also halt the larval locomotion. | *Figure 3—video 9*; *Figure 3—video 10*; *Figure 3—video 11*; *Figure 3—video 12* |
| 6B | *CG4328-p65.AD, vg-GAL4-DBD*: Activation in intact animals drove inhibition in wing buzzing and leg movements of the tethered flies. Activation in decapitated animals halted sporadic leg movements and drove a subtle change in the posture. | *Figure 3—video 14*; *Figure 3—video 15* |
| 7B | *sv-p65.AD, unc-4-GAL4-DBD*: Upon 7B activation, both decapitated and intact animals raised their wings and attempted take-offs, but only a few showed modest take-off behavior. We also observed tibia levitation in response to activation. Harris et al. observed robust take-off behavior. | *Figure 3—video 16*; *Figure 3—video 17*; *Figure 3—video 18* |
| 8A | *ey-p65.AD, ems-GAL4-DBD*: Activation brings the body of the fly closer to the ground likely flexing leg segments in both intact and decapitated animals. Harris et al. observed minimal effects after activation. | *Figure 3—video 19*; *Figure 3—video 20* |
| 8B | *C15-p65.AD, Lim3-GAL4-DBD*: Activation drove intact animals lean backward and take-off. A few animals initiated wing flapping after the jump; others failed to initiate wing flapping and fell after the jump, then they jumped again under the continuous activation. Decapitated animals showed a similar response but never initiated the wing flapping after the take-off. | *Figure 3—video 21*; *Figure 3—video 22*-2 |
| 9A | *Dr-p65.AD, Gad1-GAL4-DBD*: Activation in intact animals drove erratic forward locomotion of the animal. Activation in tethered intact flies restricted the legs to stay in a specific posture. In decapitated animals, bodies were lowered toward the ground with legs becoming more splayed for approximately two seconds before occasional forward locomotion and leg grooming, consistent with previous research by Harris et al. | *Figure 3—video 23*; *Figure 3—video 24*; *Figure 3—video 25* |
| 9B | *acj6-p65.AD, VGlut-GAL4-DBD*: Activation in intact animals did not lead to any robust behavior; occasionally animals changed their posture mildly. Decapitated animals halted their grooming in response to 9B activation. This halting behavior was less penetrant compared to the halting behavior observed with 5B activation. | *Figure 3—video 26*; *Figure 3—video 27*; |
| 10B | *Hb9-p65.AD, sens-2-GAL4-DBD*: Activation in intact animals drove erratic walking behavior. 10B activation in decapitated animals drove leg extension and body twisting. Our findings differed from *Harris et al., 2015*, which showed erratic leg movements causing backward locomotion with occasional wing flicking and buzzing. | *Figure 3—video 28*; *Figure 3—video 29*; |
| 11 A | *tey-VP16.AD, unc-4-GAL-4-DBD*: Low intensity light activation drove lateral wing waving with occasional jumping, while high intensity activation drove wing buzzing and jumping in intact and decapitated animals. | *Figure 3—video 30*; *Figure 3—video 31*; *Figure 3—video 32*; *Figure 3—video 33* |
| 11B | *eve-p65.AD, Gad1-GAL4-DBD*: Harris et al. observed take-off behavior after activation of the 11B neurons. However, upon light activation, we observed wing movements without any take-off behavior. The wings moved from side to side in a buzzing behavior. | *Figure 3—video 34*; *Figure 3—video 35* |
| 12 A | *TfAP2-p65.AD, unc-4-GAL4-DBD*: CsChrimson expression showed a lethal phenotype with no surviving adults. We generated lineage clones using TfAP-2-GAL4. Animals expressing CsChrimson in 12 A neurons in one side of the T1 segment showed a single swing movement of the leg that is located on the same side as the animal lineage clone. We also observed bilateral wing buzzing. | *Figure 3—video 36*; *Figure 3—video 37* |
| 13 A | *dmrt99B-p.65AD, dbx-GAL4-DBD*: Upon 13 A activation, intact flies halt their walking and grooming behaviors and change the body posture, making flies slightly taller due to likely femur-coxa extension. Decapitated flies also halt the grooming behavior in response to 13 A activation. Both intact and decapitated flies buzz their wings in response to activation, a phenotype likely arising from contaminating neurons. | *Figure 3—video 38*; *Figure 3—video 39* |
| 13B | *D-VP16.AD, vg-GAL4-DBD*: Intact flies lost control of their legs and fell on their back with uncoordinated leg movements upon activation of 13B neurons. Decapitated flies responded with a postural change and a weak leg extension phenotype. | *Figure 3—video 40*; *Figure 3—video 41* |

*Table 2 continued on next page*

*Table 2 continued*

| Lineage | *Genotype*: Phenotype | Videos |
|---|---|---|
| 14 A | *Dr-p65.AD, toy-GAL4-DBD*: Activation caused intact animals to fall on their back or side with uncoordinated leg movements; flies remained uncoordinated until the cessation of the stimulus. In decapitated animals, activation drove the femur-tibia joint to move anteriorly, most pronounced in the middle legs. We also observed flexion of the legs. | *Figure 3—video 42*; *Figure 3—video 43* |
| 15B | *VGlut-p65.AD, HGTX-GAL4-DBD*: Upon light stimulation in both intact and decapitated flies, the legs showed a severe flexing phenotype. The legs flexed tightly against the body with the flies falling into a fetal position until after light stimulation ended. | *Figure 3—video 44*; *Figure 3—video 45* |
| 16B | *Hb9-p65.AD, Bi-GAL4-DBD*: Activation in both intact and decapitated animals drove flexion at the femur-tibia joint and coxa-femur axis joint causing the animal to sink lower to the ground. | *Figure 3—video 46*; *Figure 3—video 47* |
| 17 A | *unc-4-p65.AD, Hmx-GAL4-DBD*: Activation of 17 A neurons drove flexion of all the leg segments in both decapitated and intact animals. | *Figure 3—video 48*; *Figure 3—video 49* |
| 19 A | *scro-p65.AD, dbx-GAL4-DBD*: Activation in decapitated animals drove flexion at the tibia-tarsus joint as well as anterior movement of the femur-tibia axis. In intact animals, we observed severe flexing of the legs against the body, making flies fall on their back. Harris et al. observed a leg-waving phenotype of the T2 legs in decapitated animals after stimulation. | *Figure 3—video 50*; *Figure 3—video 51* |
| 21 A | *Dr-p65.AD, tj-GAL4-DBD*: Activation of 21 A neurons in decapitated animals drove flexion of the legs, bringing the body of the fly closer to the ground. We observed a similar phenotype in intact animals tethered to a pin. | *Figure 3—video 52*; *Figure 3—video 53* |
| 23B | *unc-4-p65.AD, acj6-GAL4-DBD*: Activation caused intact animals to fall on their back due to uncoordinated leg movements and sustained flexion or extension of the leg segments; flies remained uncoordinated until the cessation of the stimulus. Flies also showed increased grooming activity. We also observed wing buzzing in response to activation. Decapitated animals showed similar responses. | *Figure 3—video 54*; *Figure 3—video 55* |

*Harris et al., 2015*), we observed that both decapitated and intact animals extended their legs in response to activation.

## Hemilineage 12A

Hemilineage 12 A neurons are cholinergic and excitatory and display segment-specific and complex intersegmental projections to wing and leg nerve bundles (*Marin et al., 2024*; *Lesser et al., 2024*). We used the *unc-4*-GAL4[DBD] and *TfAP2*-GAL4[AD] driver line to express CsChrimson in 12B neurons. None of these animals, however, survived to adulthood, not even in the absence of retinal, the cofactor required for CsChrimson activity. To overcome this issue, we generated stochastic FLP-based lineage clones that expressed CsChrimson in 12 A neurons in one or a few hemisegment(s). We then optogenetically activated decapitated flies and recorded their behavior (*Figure 3—video 36*, *Figure 3—video 37*), followed by dissection and immunostaining to visualize which lineage clones were responsible for the observed phenotype. We found two cases where optogenetic activation resulted in bilateral wing opening and a leg swing. The segment and side of the 12 A lineage clone corresponded to the side of the leg that moved (*Figure 6F and G*). We also observed the following behavioral phenotypes in response to optogenetic activation, but we did not dissect the animals to further identify the lineage clone: high frequency wing beating, backward walking immediately after the stimulus termination, and abdominal extension and bending. These results indicate that 12 A neurons, as expected from their complex projections, control a magnitude of behaviors.

## Hemilineage 21A

Hemilineage 21 A neurons are glutamatergic, likely inhibitory interneurons, and innervate the leg neuropil in all thoracic segments. To assess the behaviors executed by 21 A neurons, we used two different driver lines: *Dr*-GAL4[AD] and *ey*-GAL4[DBD] or *Dr*-GAL4[AD] and *tj*-GAL4[DBD]. Both combinations target most of the 21 A neurons, the latter with higher specificity, yet both lines showed consistent results upon optogenetic activation. Stimulation of either intact or decapitated animals forced the leg segments into a specific geometry (*Figure 3—video 52*, *Figure 3—video 53*). In tethered intact animals, whose legs are freely moving in the air, we observed a clear flexion in the femur-tibia joint (*Figure 6H–J*). To test whether 21 A neurons are necessary for the relative femur-tibia positioning, we eliminated 21 A neurons by expressing UAS-*hid* with *Dr*-GAL4[AD], *ey*-GAL4[DBD]. Flies lacking 21 A neurons showed aberrant walking patterns. We observed that femur-tibia joints of the hind legs

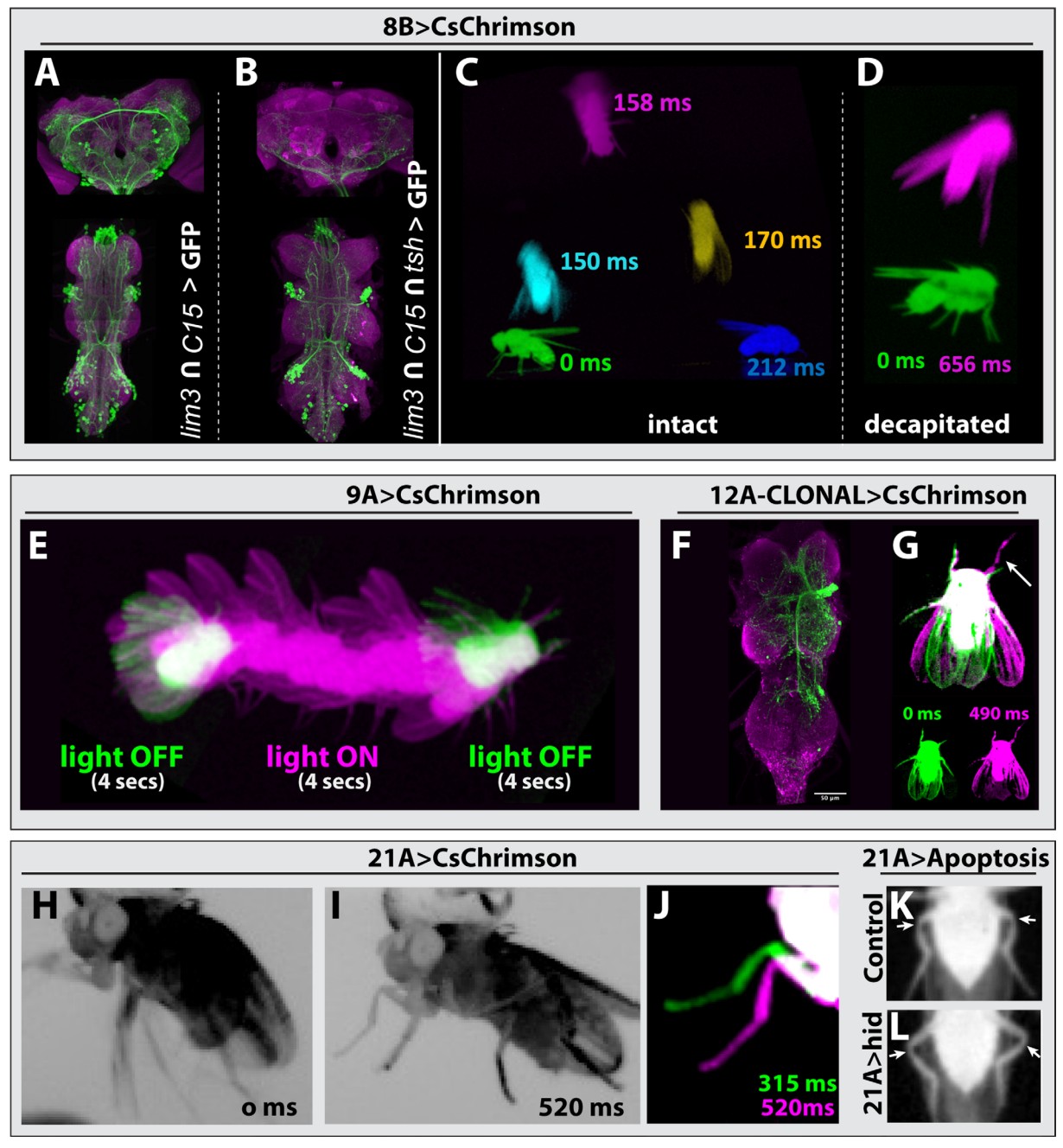

**Figure 6.** Behavioral analysis with targeted lineage manipulation. (**A–D**) Optogenetic activation of hemilineage 8 A in the VNC triggers jump behavior. *lim3*-GAL4[DBD]; *c15*-GAL4[AD]-driven CsChrimson::mVenus (green) targets 8B neurons in the VNC but also shows an unwanted broad brain expression (**A**), which can be suppressed via an additional layer of intersection using teashirt (tsh)-lexA-driven FLP strategy (**B**). (**C, D**) Overlay of video frames to capture the jump sequence induced by optogenetic activation of lineage 8B in the VNC. Intact flies (**C**) and decapitated flies (**D**) jump without raising their wings upon optogenetic activation, but decapitated flies were slower to initiate the jump. (**E**) Optogenetic activation of hemilineage 9 A induces forward walking in decapitated flies. (**F, G**) Clonal stimulation of hemilineage 12 A in the VNC in decapitated flies induces bilateral wing opening and single-step behavior. (**F**) Confocal stack displaying the lineage 12 A clone that extends from T2 into T1 and T3. (**G**) Overlay of movie frames. The fly folds both wings outward and swings its right front leg forward upon optogenetic activation. (**H, L**) Optogenetic activation of hemilineage 21 A in the VNC on a tethered, intact fly triggers flexion of the tibia-femur joint. (**H**) Without stimulus, all the legs move erratically in response to being tethered. (**I**) Upon optogenetic activation, all legs are pulled toward the body, the tibia-femur joints are flexed, and animals stay in this position until the end of stimulus. (**J**) Overlay of the movie shown in panel H and I, zoomed in on the left T1 leg. Note how the leg is pulled towards the body upon activation (520 ms) compared to its more lateral position without activation (315 ms). (**K, L**) Elimination of 21 A neurons makes hind leg femur-tibia joints protrude laterally (**L**) compared to control animals (**K**). For all overlays of movies, green display frames without optogenetic activation, magenta with optogenetic activation.

*Figure 6 continued on next page*

*Figure 6 continued*

The online version of this article includes the following figure supplement(s) for figure 6:

**Figure supplement 1.** Giant fiber (GF) connectome.

protruded laterally compared to the control sibling flies (*Figure 6*, K, L). Our results showed that 21 A neurons control the relative positioning of the leg segments, especially the femur and tibia.

## Other anatomical region of interest

While characterizing the expression pattern of the gene-specific split-Gal4 library, we noted that the applicability of these tools extends beyond the VNC. A total of 24 driver lines targeted clusters of neurons in the subesophageal zone (SEZ) (*Table 1*). The SEZ processes mechanosensory and gustatory sensory input and controls motor output related to feeding behavior. It is anatomically part of the VNC and comprises the first three segments of the VNC, which are populated by NBs that are the segmental homologs of NBs found in the thoracic and abdominal segments of the VNC (*Doe and Goodman, 1985*; *Kendroud et al., 2018*; *Li et al., 2014*). A key difference is that only a small number of NBs pairs survive in the SEZ (*Kuert et al., 2014*). The SEZ NBs are expected to express a similar set of transcription factors as their thoracic counterparts. Therefore, these transcription factors and their corresponding split-GAL4 driver lines are excellent tools to target and manipulate homologous lineages in the SEZ.

## Discussion

The ability to trace neuronal lineages across their developmental journey and to manipulate their function is essential to investigate how neurons interconnect to form neuronal circuits and regulate specific behaviors. To address these questions, most studies have focused on a few specific regions of the CNS, for example, the mushroom body in flies, for which specific genetic tools exist to target defined neuronal populations during developmental and adult life. In this study, we utilized scRNAseq data from the VNC (*Allen et al., 2020*) that completed its annotation for all but one hemilineage and analyzed the transcriptome of individual hemilineages. Through this effort, we identified new marker genes for hemilineages, verified their expression patterns in the VNC, and created split-GAL4 driver lines for 24 lineage-specific marker genes lines by editing their genomic loci. By employing binary combinations of these new lines amongst each other or with driver lines established previously, we constructed a comprehensive split-GAL4 library that targets 32 out of 34 hemilineages during development and adult life (*Table 1*), enabling the genetic dissection of how each hemilineage contributes to circuit development (*Lacin et al., 2020*; *Lacin et al., 2019*; *Lacin et al., 2024*; *Chen et al., 2023a*; *Xie et al., 2021*; *Xie et al., 2019*).

### Mapping and manipulating morphological outgrowth patterns of hemilineages during development

The driver line combinations presented here are developmentally stable, and most combinations label both embryonic and post-embryonic neurons of the target hemilineage. This makes them a valuable resource for lineage-based dissection of larval nervous system development and function. Furthermore, they target individual hemilineages throughout metamorphosis and adult life. This is critical as the formation and maturation of adult neuronal circuits take place during metamorphosis and last several days. This time window is greatly prolonged compared to the rapid development of larval circuits that occurs within a few hours during embryogenesis and offers greater opportunities for experimental manipulation. By layering temperature or light-controlled genetic effectors, like shibere[TS], channelrhodopsins, or LACE-Cas9 (*Polstein and Gersbach, 2015*; *Mohammad et al., 2017*) with our toolkit, researchers can manipulate neuronal and gene activity with high temporal resolution. This makes it possible to investigate dynamic processes such as synapse formation, circuit assembly, and functional maturation. For example, recent studies demonstrated that developing neuronal circuits exhibit patterned calcium activities during metamorphosis, and these activities likely regulate the synaptic connectivity among neurons (*Akin et al., 2019*; *Bajar et al., 2022*). Shibere[TS] expression driven by our driver lines can inhibit the developmentally observed neuronal activity in a specific

hemilineage during a specific time window to test whether this manipulation alters the synaptic connectivity of the hemilineage.

## 8B neurons likely function in the giant-fiber escape circuit

Our work here demonstrated that the activation of 8B neurons elicits robust take-off behavior that closely resembles the GF-induced take-off response (*Card and Dickinson, 2008*). This observation raises an intriguing question: do 8B neurons function in the GF escape circuit? Interestingly, although 8B neurons do not appear to connect directly to TTMns, the primary output neurons of the escape circuit (*Marin et al., 2024*), we report that they do form a complex synaptic relationship with the GF. Specifically, a subset of 8B neurons is both upstream and downstream synaptic partners of the GF, accounting for 25% of GF's synaptic input and 12.5% of GF's output. This synaptic loop centered around the GF neurons suggests a recurrent feedback mechanism within the GF circuit. Given that hemilineage 8B neurons exhibit interconnectivity with each other and receive leg proprioceptive input (*Marin et al., 2024*), we speculate that lineage 8B may function as an integrator in and amplifier of the GF circuit. This example underscores that our split-GAL4 library provides an excellent resource for further exploration of lineage-coupled behavior.

## Addressing lineage differentiation by studying cell heterogeneity within hemilineages

Our lineage annotation of the VNC transcriptome revealed that most hemilineages are represented by more than one RNAseq cluster, which reflects heterogeneity within a hemilineage and indicates that hemilineages can be further subdivided into subclasses of neurons. Indeed, we found that such subclasses express specific transcription factors, which can be considered subclass-defining factors, for example Tj in hemilineage 0 A (*Figure 2F*). The tools we present here form a starting point to visualize or manipulate neuronal subclasses within a hemilineage. For example, one can use the split-GAL4 driver line combinations to express an UAS transgene preceded by an FRT-flanked stop codon in a specific lineage. Flippase expression can be easily restricted to a subclass of neurons in a hemilineage with the LexA/LexAop system under control of the subclass-defining transcription factor. As a result, the transgene will only be expressed in a subclass of neurons in a hemilineage. Instead of working with subclass-defining transcription factors, one can also use birth-order marking temporal genes such as *chinmo*, *mamo*, or *broad-c* (*Liu et al., 2019*; *Maurange et al., 2008*; *Zhu et al., 2006*; *Zhou et al., 2009*) to restrict driver activity to a group of neurons born in a specific temporal window within a hemilineage. Thus, with a strategic combination of orthogonal gene-specific driver system (e.g. split-GAL4, LexA, and QF), one can now dissect the neuronal circuit formation with unprecedented precision.

In conclusion, our study underscores the potential of temporally stable driver lines to target hemilineages in the VNC during development and adult life. This approach enables future studies investigating how neurons acquire their specific fates and integrate into the broader networks of neural networks that control intricate animal behaviors.

## Materials and methods
### scRNA-seq data analysis

Candidate genes to convert into split-GAL4 driver lines were identified using the scRNA-seq data generated by *Allen et al., 2020*. No modifications were made to the preprocessing pipeline, and we used the cluster markers defined by *Allen et al., 2020* and investigated the combinatorial expression patterns of the highest expressed cluster markers in binary combinations using the code shared on GitHub using Seurat v5 (https://github.com/aaron-allen/VNC_scRNAseq; *Allen, 2021*). Candidate genes to make split-GAL4 drivers from were chosen based on their ability to selectively mark the scRNAseq cluster(s) covering a hemilineage and selected markers that were expressed by near all cells of the hemilineage for the downstream experimental validation steps. We then prioritized testing these combinations based on the availability of antibodies, BAC lines, and CRiMIC/MiMIC constructs to validate their expression pattern prior to creating split-GAL4 lines for these candidates.

## Fly stocks

Fly stocks were reared on the standard cornmeal fly food at 25 °C unless indicated otherwise. Fly lines used in this study are listed in the Key Resources Table. A current inventory of gene-specific split-GAL-4 lines is maintained by Yu-Chieh David Chen and Yen-Chung Chen from Claude Desplan's lab (https://www.splitgal4.org). Lines were contributed by the labs of Claude Desplan, Liqun Lue, Benjamin White, Norbert Perrimon, and Haluk Lacin's laboratories. Behavior was tested at room temperature (22–25°C) 2–10 days post-eclosion. Genotypes of animals used in figures and videos are shown in *Supplementary file 5*.

## Clonal analysis

Wild type MARCM analysis was performed as described before (*Lee and Luo, 1999*). Animals were heat-shocked within 24 hr after egg hatching (*Lacin et al., 2014*). Multi-Color FLP-Out NB3-5 (lineage 9) clones were generated with 49C03-GAL4 crossed to hsFlp2::PEST;; HA_V5_FLAG as described before (*Lacin and Truman, 2016*; *Nern et al., 2015*). 20X-UAS>dsFRT > CsChrimson mVenus_attp18, hs-Flp2PESt_attp3 X Tf-AP2-GAL4: lineage clones were generated via heat-shock within 24 hours window after egg hatching.

## Gene editing

### Introduction of Trojan split-GAL4 by recombinase-mediated cassette exchange

Gene-specific split-GAL4$^{AD}$ and split-GAL4$^{DBD}$ lines were made from MiMIC or CRIMIC lines via Trojan exon insertion as described before (*Lacin et al., 2019*; *Chen et al., 2023a*; *Diao et al., 2015*; *Nagarkar-Jaiswal et al., 2015*). Briefly, pBS-KS-attB2-SA(0,1, or 2)-T2A-Gal4DBD-Hsp70 or pBS-KS-attB2-SA(0,1, or 2)-T2A-p65AD-Hsp70 were co-injected with phiC31 integrase into the respective MiMIC/CRIMIC parent stock (Key Resources Table). Transformants were identified via the absence of y+or 3xP3-GFP markers. The correct orientation of the construct was validated by GFP signal upon crossing the putative hemidriver to a line carrying the counter hemidriver under control of the tubulin promoter and an UAS-GFP transgene (Key Resources Table).

### Insertion of gene-specific Trojan split-GAL4 construct with CRISPR

Guide RNAs (gRNA) were selected to target all expressed isoforms in an amendable intronic region or to the 3' end of the gene if no suitable intron was present (e.g. *fer3 and ems*; Key Resources Table, *Supplementary file 4*). gRNAs were identified with CRISPR target Finder for vas-Cas9 flies, BDSC#51324 with maximum stringency and minimal off-target effects (*Gratz et al., 2014*). gRNA targeting *hb9, vg,* and *H15* was cloned into pCFD4 together with a guide RNA to linearize the donor vector (*Port et al., 2014*; *Kanca et al., 2019*). The remainder of the guides was synthesized into pUC57_GW_OK2 (Genewiz/Azenta (Burlington, MA)).

CRISPR donors were generated using a modified version of the strategy developed by *Kanca et al., 2022*. We used the Genewiz company to synthesize a DNA fragment into the EcoRV site of the pUC57-GW- OK2 vector. This fragment is made of the left and right homology arms (HA) which are immediately adjacent to the gRNA cut site and restriction enzyme sites (SacI-KpnI) between these arms (*Figure 3—figure supplement 2A*). We then directionally cloned the Sac1-attP-FRT-splitGAL4-FRT-attP-KpnI fragment (*Figure 3—figure supplement 2B*) in between the left and right HAs using the SacI and KpnI sites. Note that SacI and Kpn should only be chosen when the homology arms do not have these cut sites. To facilitate this last step, we generated universal plasmids in each reading frame for each hemi driver, DBD and p65.AD in the original Trojan vector backbones, referred to as pBS-KS-attP2FRT2-SA-T2AGAL4[AD or DBD (0,1,2)]-Hsp70 with Gibson assembly, combining the following fragments:

1. pBS-KS backbone from the original Trojan vector (digested with SacI and KpnI).
2. the exon (consisting of splice acceptor, GAL4-DBD or p65.AD, and Hsp70 Poly A signal) was PCR-amplified from the original Trojan vectors (e.g. pBS-KS-attB2-SA(0)-T2A-p65AD-Hsp70) with the following primers:

F: 5' ctagaaagtataggaacttcGAATTCagtcgatccaacatggcgacttg 3'

R:5' ctttctagagaataggaacttcGATATCaaacgagtttttaagcaaactcactcc 3

Note EcoRI and EcoRV (capitalized) sites were included as a back-up strategy for replacing the Trojan exon between attP FRT if needed.

5' SacI-attP-FRT sequence was PCR amplified from pM14 (*Kanca et al., 2022*) with primers:
F: 5' actcactatagggcgaattgGAGCTCacggacacaccgaag 3'
R: 5' caagtcgccatgttggatcgac *3'*

3' FRT- attP-KpnI sequence PCR amplified from pM14 (*Kanca et al., 2022*) with primers:

F: 5' ggagtgagtttgcttaaaaactcgtttGATATCgaagttcctattctctagaaag 3'
R: 5' cactaaagggaacaaaagctgggtaccgtactgacggacacaccgaag *3'*

Corresponding sequences from pBS-KS are underlined, pM14 are in italics, and Trojan AD/DBD are in bold; restriction enzyme sites are in all caps. All plasmids were validated by Sanger sequencing (Genewiz/Azenta Burlington, MA).

Note that for *hb9, vg, sens-2, H15, scro, Ets21C* and *eve* we inserted the T2A- split-GAL4$^{DBD}$ and/or T2A-split-GAL4$^{p65-AD}$ into the host gene intron as a Trojan exon with flanking FRT sites in a similar manner to CRIMIC lines generated by the Bellen Lab (detailed below). However, since this is problematic for FLP-dependent mosaic experiments, we generated additional lines for *hb9, sens2, Ets21C eve* and *vg* lacking FRT sites by replacing the FRT-flanked cassettes with the original White lab Trojan AD/DBD exons via attP-phiC31-mediated recombination as described above.

Split-GAL4 drivers for and D were made by the Erclick laboratory. CRISPR-mediated gene editing was performed by WellGenetics Inc using modified methods of *Kondo and Ueda, 2013*. For *fkh*, the gRNA sequence GTGACATCACCAATACCCGC[TGG] was cloned into a U6 promoter plasmid. Cassette T2A-Gal4DBD-RFP, which contains T2A, Gal4DBD, a floxed 3xP3-RFP, a Hsp70Ba 3'UTR, and two homology arms, was cloned into pUC57-Kan as donor template for repair. *fkh/CG10002*-targeting gRNAs and hs-Cas9 were supplied in DNA plasmids, together with donor plasmid for microinjection into embryos of control strain w[1118]. F1 flies carrying the selection marker of 3xP3-RFP were further validated by genomic PCR and sequencing. CRISPR generates a break in fkh/CG10002 and is replaced by cassette T2A-Gal4DBD-RFP.

Similarly, for *D*, the gRNA sequences ACTCGACTCTAATAGAGCAC[CGG] /GCACCGGAACCGGTCGCCTC[AGG] were cloned into U6 promoter plasmid(s). Cassette T2A-VP16AD-3XP3-RFP, which contains T2A, VP16AD, and a floxed 3xP3-RFP, and two homology arms were cloned into pUC57-Kan as donor template for repair. *D/CG5893*-targeting gRNAs and hs-Cas9 were supplied in DNA plasmids, together with donor plasmid for microinjection into embryos of control strain w[1118]. F1 flies carrying the selection marker of 3xP3-RFP were further validated by genomic PCR and sequencing. CRISPR generates a break in *D/CG5893* and is replaced by cassette T2A-VP16AD-3XP3-RFP.

## Direct split-GAL4 insertion with CRISPR

For *fer3, ems*, and *HLH4C*, we inserted T2A-GAL4$^{DBD}$ directly in frame with the last coding exon instead of inserting it into an intron as a Trojan exon flanked by attP and FRT sites. The gRNA and entire donor region (a LHA-GAL4-DBD-RHA fragment, without attP and FRT sequences) were synthesized in pUC57_gw_OK2 and injected into vas-Cas9 flies (w[1118]; PBac(y[+mDint2]=vas-Cas9)VK00027) by Rainbow transgenics (Camarillo, CA). Transformed animals were crossed to flies carrying Tubulin-GAL4$^{AD}$, UAS-TdTomato, and offspring was scored for TdTomato expression to identify positive lines. The expression pattern of the reporter served as a verification for correct editing events; no further verification was performed.

## Immunochemistry and data acquisition

Samples were dissected in phosphate buffered saline (PBS) and fixed with 2% paraformaldehyde in PBS for an hour at room temperature and then washed several times in PBS-TX (PBS with 1% Triton-X100) for a total of 20 min. Tissues were incubated with primary antibodies (Key Resources Table) for two to four hours at room temperature or overnight at 4 °C. After three to four rinses with PBS-TX to remove the primary antisera, tissues were washed with PBS-TX for an hour. After wash, tissues secondary antibodies were applied for 2 hr at room temperature or overnight at 4 °C. Tissues were washed again with PBS-TX for an hour and mounted in Vectashield or in DPX after dehydration

through an ethanol series and clearing in xylene (*Truman et al., 2004*). Images were collected with 20 X or 40 X objectives using confocal microscopy. Images were processed with Image J/FIJI.

### Behavioral analysis

For optogenetic stimulation, we used standard food containing 0.2 mM all-trans retinal. As a light source for optogenetic activation, we used either white light coming from the gooseneck guide attached to the halogen light box or red light (Amazon-Chanzon, 50 W, Led chip, 620 nm - 625 nm / 3500 - 4000LM). Animal behaviors were recorded via a USB-based Basler Camera (acA640-750um) under continuous infrared light source (Amazon-DI20 IR Illuminator).

## Acknowledgements

We thank the Lacin laboratory members for critical reading of the manuscript, discussion and suggestions. We thank Aaron Allen and Stephen Goodwin for sharing their code for scRNAseq data analysis and sharing their fly lines prior to publication and Dorothea Godt, Angelike Stathopoulos and Gerald Campbell for gifting antibodies. We also thank Hugo Bellen and Oguz Kanca for sharing their reagents. Many stocks obtained from the Bloomington *Drosophila* Stock Center (NIH P40OD018537) were used in this study as well as antibodies from the Developmental Studies Hybridoma Bank, created by the NICHD of the NIH and maintained at the University of Iowa, Department of Biology, Iowa City, IA 52242. This work was supported by grants from the National Institutes of Health to J.B.S. (R01NS036570), and to H.L. (R01NS122903), and by funding from HHMI to J.W.T.

## Additional information

### Funding

| Funder | Grant reference number | Author |
|---|---|---|
| NIH Office of the Director | R01NS122903 | Haluk Lacin |
| NIH Office of the Director | R01NS036570 | James B Skeath |

The funders had no role in study design, data collection and interpretation, or the decision to submit the work for publication.

### Author contributions

Jelly HM Soffers, Conceptualization, Data curation, Validation, Investigation, Visualization, Methodology, Writing – original draft, Writing – review and editing; Erin Beck, Validation, Visualization, Methodology; Daniel J Sytkowski, Marianne E Maughan, Methodology, Writing – review and editing; Devasri Devarakonda, Beth A Wilson, Methodology; Yi Zhu, Conceptualization, Methodology; Yu-Chieh David Chen, Resources, Writing – review and editing; Ted Erclik, Resources; James W Truman, James B Skeath, Funding acquisition, Writing – review and editing; Haluk Lacin, Conceptualization, Data curation, Supervision, Funding acquisition, Validation, Investigation, Visualization, Methodology, Writing – original draft, Project administration, Writing – review and editing

### Author ORCIDs

Jelly HM Soffers ⓘ http://orcid.org/0000-0002-1051-7375
Yu-Chieh David Chen ⓘ https://orcid.org/0000-0002-2597-7577
James W Truman ⓘ https://orcid.org/0000-0002-9209-5435
James B Skeath ⓘ https://orcid.org/0000-0003-1179-4857
Haluk Lacin ⓘ https://orcid.org/0000-0003-2468-9618

Reviewer #1 (Public review): https://doi.org/10.7554/eLife.106042.3.sa1
Reviewer #2 (Public review): https://doi.org/10.7554/eLife.106042.3.sa2
Reviewer #3 (Public review): https://doi.org/10.7554/eLife.106042.3.sa3
Author response https://doi.org/10.7554/eLife.106042.3.sa4

## Additional files

### Supplementary files
Supplementary file 1. Detailed description of the expression patterns of the driver lines used in *Figure 3*, *Figure 3—figure supplement 1*.

Supplementary file 2. Synaptic inputs of the Giant Fiber neuron related to *Figure 6*.

Supplementary file 3. Synaptic outputs of the Giant Fiber neuron related to *Figure 6*.

Supplementary file 4. Additional information on CRISPR genomic edits.

Supplementary file 5. Genotypes of animals used for each figure and video.

MDAR checklist

### Data availability
All data generated or analysed during this study are included in the manuscript and supporting files.

The following previously published dataset was used:

| Author(s) | Year | Dataset title | Dataset URL | Database and Identifier |
|---|---|---|---|---|
| Allen AM, Neville MC, Birtles S, Croset V, Treiber CD, Waddell S, Goodwin SF | 2020 | A single-cell transcriptomic atlas of the adult *Drosophila* ventral nerve cord | https://www.ncbi.nlm.nih.gov/geo/query/acc.cgi?acc=GSE141807 | NCBI Gene Expression Omnibus, GSE141807 |

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

# Appendix 1

**Appendix 1—key resources table**

| Reagent type (species) or resource | Designation | Source or reference | Identifiers | Additional information |
|---|---|---|---|---|
| Antibody | guinea pig anti-tj polyclonal | Gift from Dorothea Godt | | 1:5000 dilution |
| Antibody | Rabbit anti-tey polyclonal | Gift from Angelike Stathopoulos | | 1:200 dilution |
| Antibody | Rat anti-c15 polyclonal | Gift from Gerard Campbell | | 1:1000 dilution |
| Antibody | Chicken anti-GFP polyclonal | Life Technologies | A-10262 | 1:1000 dilution |
| Antibody | Rabbit anti-GFP polyclonal | Life Technologies | A-11122 | 1:1000 dilution |
| Antibody | Rabbit anti-Unc-4 polyclonal | *Lacin et al., 2014* | A-10262 | 1:1000 dilution |
| Antibody | Mouse anti-Acj6 monoclonal | DSHB | Acj6 | 1:100 dilution |
| Antibody | Rat anti-CadN monoclonal | DSHB | DN-Ex #8 | 1:25 dilution |
| Antibody | Mouse anti-Neuroglian monoclonal | DSHB | BP104 | 1:25 dilution |
| Antibody | Goat anti-rabbit Alexa Fluor 488 | Life Technologies | A-11034 | 1:500 dilution |
| Antibody | Goat anti-rabbit Alexa Fluor 568 | Life Technologies | A-11011 | 1:500 dilution |
| Antibody | Goat anti-rabbit Alexa Fluor 633 | Life Technologies | A-21070 | 1:500 dilution |
| Antibody | Goat anti-rat Alexa Fluor 633 | Life Technologies | A-21094 | 1:500 dilution |
| Antibody | Goat anti-chicken Alexa Fluor 488 | Life Technologies | A-11039 | 1:500 dilution |
| Antibody | Goat anti-mouse Alexa Fluor 568 | Life Technologies | A-11001 | 1:500 dilution |
| Antibody | Goat anti-mouse Alexa Fluor 633 | Life Technologies | A-21050 | 1:500 dilution |
| Antibody | Goat anti-rat Alexa Fluor 568 | Life Technologies | A-21050 | 1:500 dilution |
| Genetic reagent (*D. melanogaster*) | unc-4DBD/FM7GFP; 20XUASCsChrimson_attp40/cyo | *Lacin et al., 2020* | | |
| Genetic reagent (*D. melanogaster*) | unc-4AD/FM7; 20X-UASChrimson_attp40/cyo | *Lacin et al., 2020* | | |
| Genetic reagent (*D. melanogaster*) | sens2-GAL4-DBD | *Lacin et al., 2024* | | |
| Genetic reagent (*D. melanogaster*) | P{w[+mW.hs]=GawB}elav[C155]; P{w[+mW.hs]=FRT(w[hs])}G13 P{w[+mC]=tubP GAL80}LL2 | Tzumin Lee Lab | | |
| Genetic reagent (*D. melanogaster*) | 20XUAS-CsChrimson-mVenus_attp18 | V. Jayaraman lab | | |
| Genetic reagent (*D. melanogaster*) | 20XUAS >FRT-stop>CsChrimson-mVenus_attp18 | V. Jayaraman lab | | |
| Genetic reagent (*D. melanogaster*) | P{GawB}elav[C155], P{FRT(w[hs])}G13 P{UAS-mCD8::GFP.L}LL5 | Tzumin Lee Lab | | |
| Genetic reagent (*D. melanogaster*) | P{FRT(w[hs])}G13 P{tubP-GAL80}LL2 | Tzumin Lee Lab | | |
| Genetic reagent (*D. melanogaster*) | y[1] w1118; P{tubP-GAL80}LL9 P{FRT(w[hs])}2 A/TM3, Sb | Tzumin Lee Lab | | |
| Genetic reagent (*D. melanogaster*) | knot-p65.AD/CyO, weep; Dr/TM6 | Luo Lab, Hongjie Li | | |
| Genetic reagent (*D. melanogaster*) | pin/cyo; c15-p65.AD/TM6b | Luo Lab, Hongjie Li | | |

*Appendix 1 Continued on next page*

*Appendix 1 Continued*

| Reagent type (species) or resource | Designation | Source or reference | Identifiers | Additional information |
|---|---|---|---|---|
| Genetic reagent (*D. melanogaster*) | tj-vp16.AD | Desplan Lab- David Chen | | |
| Genetic reagent (*D. melanogaster*) | twit-p65.AD | Stephen Goodwin | | |
| Genetic reagent (*D. melanogaster*) | 13XLexAop2-IVS-myr::GFP in attP40 | BDSC | RRID:BDSC32210 | |
| Genetic reagent (*D. melanogaster*) | P{hsFLP}1; P{FRT(w[hs])}G13 P{tubP-GAL80}LL2/CyO | BDSC | RRID:BDSC5145 | |
| Genetic reagent (*D. melanogaster*) | P{tubP-GAL80}LL10 P{neoFRT}40 A/CyO | BDSC | RRID:BDSC5192 | |
| Genetic reagent (*D. melanogaster*) | w[*]; l(2)*[*]/CyO; Mi{Trojan-GAL4DBD.0}ChAT[MI04508-TG4DBD.0] CG7715[MI04508-TG4DBD.0-X]/TM3, Sb[1] | BDSC | RRID:BDSC60318 | |
| Genetic reagent (*D. melanogaster*) | w1118; PBac{RB}Fer2e03248 | BDSC | RRID:BDSC26028 | |
| Genetic reagent (*D. melanogaster*) | w1118; PBac{Sp1-EGFP.S}VK00033 | BDSC | RRID:BDSC38669 | |
| Genetic reagent (*D. melanogaster*) | w[1118]; PBac{y[+mDint2] w[+mC]=fkh GFP.FPTB} VK00037/SM5 | BDSC | RRID:BDSC43951 | |
| Genetic reagent (*D. melanogaster*) | y[1] w[*]; Mi{y[+mDint2]=MIC}twit[MI06552]/(SM6a) | BDSC | RRID:BDSC41449 | |
| Genetic reagent (*D. melanogaster*) | w[*]; Mi{Trojan-GAL4DBD.0}Dbx[MI05316-TG4DBD.0]/TM6B, Tb[1] | *Lacin et al., 2019* | RRID:BDSC82989 | |
| Genetic reagent (*D. melanogaster*) | y[1] w[*]; Mi{Trojan-GAL4DBD.1}Lim3[MI03817-TG4DBD.1]/(CyO) | *Lacin et al., 2019* | RRID:BDSC82990 | |
| Genetic reagent (*D. melanogaster*) | w1118; PBac{WH}Ets21Cf03639 | BDSC | RRID:BDSC18678 | |
| Genetic reagent (*D. melanogaster*) | w[*]; Mi{Trojan-p65AD.2}VGlut[MI04979-Tp65AD.2]/CyO | *Lacin et al., 2019* | RRID:BDSC82986 | |
| Genetic reagent (*D. melanogaster*) | w[*]; betaTub60D[Pin-1]/CyO; Mi{Trojan-p65AD.1}Dr[MI14348-Tp65AD.1] | *Lacin et al., 2019* | RRID:BDSC82991 | |
| Genetic reagent (*D. melanogaster*) | y[1] w[*]; Mi{y[+mDint2]=MIC}Dr[MI14348]/TM3, Sb[1] Ser[1] | BDSC | RRID:BDSC59504 | |
| Genetic reagent (*D. melanogaster*) | w[*]; betaTub60D[Pin-1]/CyO; TI{2 A-GAL4(DBD)::Zip-}HGTX[DBD]/TM6B, Tb[1] | *Lacin et al., 2019* | RRID:BDSC82992 | |
| Genetic reagent (*D. melanogaster*) | ey-GAL4-DBD | *Lacin et al., 2019* | RRID:BDSC6294 | |
| Genetic reagent (*D. melanogaster*) | y[1] w[*]; Mi{y[+mDint2]=MIC}Ets65A[MI05707] | BDSC | RRID:BDSC40235 | |
| Genetic reagent (*D. melanogaster*) | y[1] w[*]; TI{GFP[3xP3.cLa]=CRIMIC.TG4.2}sv[CR00370-TG4.2] | BDSC | RRID:BDSC78901 | |
| Genetic reagent (*D. melanogaster*) | y[1] w[*]; TI{GFP[3xP3.cLa]=CRIMIC.TG4.2} Sox21a[CR00451-TG4.2]/TM3 Sb[1] Ser[1] | BDSC | RRID:BDSC83174 | |
| Genetic reagent (*D. melanogaster*) | y[1] w[*] Mi{y[+mDint2]=MIC}bi[MI08152] lncRNA:CR32773[MI08152] | BDSC | RRID:BDSC51220 | |
| Genetic reagent (*D. melanogaster*) | y[1] w[*]; Mi{y[+mDint2]=MIC}ap[MI01996]/CyO | BDSC | RRID:BDSC42297 | |
| Genetic reagent (*D. melanogaster*) | y[1] w[*]; Mi{y[+mDint2]=MIC}inv[MI09433] | BDSC | RRID:BDSC52163 | |

*Appendix 1 Continued on next page*

*Appendix 1 Continued*

| Reagent type (species) or resource | Designation | Source or reference | Identifiers | Additional information |
|---|---|---|---|---|
| Genetic reagent (*D. melanogaster*) | y[1] w[*] Mi{y[+mDint2]=MIC}acj6[MI07818] | BDSC | RRID:BDSC51212 | |
| Genetic reagent (*D. melanogaster*) | y[1] w[*]; Mi{PT-GFSTF.2}Hmx[MI02025-GFSTF.2]/TM3, Sb[1] Ser[1] | BDSC | RRID:BDSC59785 | |
| Genetic reagent (*D. melanogaster*) | y[1] w[*]; Mi{y[+mDint2]=MIC}Hmx[MI02896] | BDSC | RRID:BDSC36161 | |
| Genetic reagent (*D. melanogaster*) | y[1] w[*]; Mi{y[+mDint2]=MIC}Ets65A[MI05707] | BDSC | RRID:BDSC40235 | |
| Genetic reagent (*D. melanogaster*) | y[1]; Mi{y[+mDint2]=MIC}toy[MI03240] | BDSC | RRID:BDSC61701 | |
| Genetic reagent (*D. melanogaster*) | P{Tub-dVP16AD.D} | BDSC | RRID:BDSC60295 | |
| Genetic reagent (*D. melanogaster*) | P{Tub-GAL4DBD.D} | BDSC | RRID:BDSC0298 | |
| Genetic reagent (*D. melanogaster*) | lim3-GAL4-DBD | *Lacin et al., 2019* | RRID:BDSC82990 | |
| Genetic reagent (*D. melanogaster*) | ChAT-p65.AD | *Lacin et al., 2019* | | RMCE with RRID:BDSC37817 |
| Genetic reagent (*D. melanogaster*) | y[*]w[*]/w[*];inv[MI09433.p65AD_1]/SM6a | this study | | RMCE with RRID:BDSC52163, request from Lacin lab |
| Genetic reagent (*D. melanogaster*) | ap-GAL4-DBD | this study | | RMCE with RRID:BDSC42297, request from Lacin lab BDSC52163 |
| Genetic reagent (*D. melanogaster*) | ap-p65.AD | this study | | RMCE with RRID:BDSC42297, request from Lacin lab |
| Genetic reagent (*D. melanogaster*) | mab-21-GAL4-DBD | this study | | RMCE with RRID:BDSC59220, request from Lacin lab |
| Genetic reagent (*D. melanogaster*) | mab-21-p65.AD | this study | | RMCE with RRID:BDSC59220, request from Lacin lab |
| Genetic reagent (*D. melanogaster*) | toy-GAL4-DBD | this study | | RMCE with RRID:BDSC61701, request from Lacin lab |
| Genetic reagent (*D. melanogaster*) | toy-p65.AD | this study | | RMCE with RRID:BDSC61701, request from Lacin lab |
| Genetic reagent (*D. melanogaster*) | shaven-p65.AD | this study | | RMCE with RRID:BDSC78901, request from Lacin lab |
| Genetic reagent (*D. melanogaster*) | sox21a-GAL4-DBD | this study | | RMCE with RRID:BDSC93174, request from Lacin lab |
| Genetic reagent (*D. melanogaster*) | bi-GAL4-DBD | this study | | RMCE with RRID:BDSC51220, request from Lacin lab |
| Genetic reagent (*D. melanogaster*) | bi-p65.AD | this study | | RMCE with RRID:BDSC51220, request from Lacin lab |

*Appendix 1 Continued on next page*

*Appendix 1 Continued*

| Reagent type (species) or resource | Designation | Source or reference | Identifiers | Additional information |
|---|---|---|---|---|
| Genetic reagent (*D. melanogaster*) | CG4328-p65.AD | this study | | RMCE with RRID:BDSC42307, request from Lacin lab |
| Genetic reagent (*D. melanogaster*) | **Ets65A-GAL4-DBD** | this study | | RMCE with RRID:BDSC56352, request from Lacin lab |
| Genetic reagent (*D. melanogaster*) | Hmx-GAL4-DBD | this study | | RMCE with RRID:BDSC36161, request from Lacin lab |
| Genetic reagent (*D. melanogaster*) | dmrt99b-GAL4-DBD | this study | | RMCE with RRID:BDSC92707, request from Lacin lab |
| Genetic reagent (*D. melanogaster*) | dmrt99b-p65.AD | this study | | RMCE with RRID:BDSC92707, request from Lacin lab |
| Genetic reagent (*D. melanogaster*) | Dr-GAL4-DBD | this study | | RMCE with RRID:BDSC59504, request from Lacin lab |
| Genetic reagent (*D. melanogaster*) | exex-GAL4-DBD | this study | | RMCE with exex-p65AD[attP2FRT2], request from Lacin lab |
| Genetic reagent (*D. melanogaster*) | vg-GAL4-DBD | this study | | RMCE with vg-p65AD[attP2FRT2], request from Lacin lab |
| Genetic reagent (*D. melanogaster*) | sens2-p65.AD | this study | | RMCE with sens2-GAL4-DBD[attP2FRT2], request from Lacin lab |
| Genetic reagent (*D. melanogaster*) | Ets21C-GAL4-DBD | this study | | RMCE with Ets21C-p65.AD[attP2FRT2], request from Lacin lab |
| Genetic reagent (*D. melanogaster*) | eve-GAL4-DBD | this study | | RMCE with eve-p65.AD[attP2FRT2], request from Lacin lab |
| Genetic reagent (*D. melanogaster*) | exex-p65.AD[attP2FRT2] | this study | | CRISPR /Trojan (CRIMIC), request from Lacin lab |
| Genetic reagent (*D. melanogaster*) | exex-GAL4-DBD[attP2FRT2] | this study | | CRISPR /Trojan (CRIMIC), request from Lacin lab |
| Genetic reagent (*D. melanogaster*) | eve-p65.AD[attP2FRT2] | this study | | CRISPR /Trojan (CRIMIC), request from Lacin lab |
| Genetic reagent (*D. melanogaster*) | vg-p65.AD[attP2FRT2] | this study | | CRISPR /Trojan (CRIMIC), request from Lacin lab |
| Genetic reagent (*D. melanogaster*) | vg-GAL4-DBD[attP2FRT2] | this study | | CRISPR /Trojan (CRIMIC), request from Lacin lab |
| Genetic reagent (*D. melanogaster*) | H15-p65.AD[attP2FRT2] | this study | | CRISPR /Trojan (CRIMIC), request from Lacin lab |

*Appendix 1 Continued on next page*

*Appendix 1 Continued*

| Reagent type (species) or resource | Designation | Source or reference | Identifiers | Additional information |
|---|---|---|---|---|
| Genetic reagent (*D. melanogaster*) | scro-p65.AD[attP2FRT2] | this study | | CRISPR /Trojan (CRIMIC), request from Lacin lab |
| Genetic reagent (*D. melanogaster*) | scro-GAL4-DBD[attP2FRT2] | this study | | CRISPR /Trojan (CRIMIC), request from Lacin lab |
| Genetic reagent (*D. melanogaster*) | Ets21C-p65.AD[attP2FRT2] | this study | | CRISPR /Trojan (CRIMIC), request from Lacin lab |
| Genetic reagent (*D. melanogaster*) | Ets21C-GAL4-DBD[attP2FRT2] | this study | | CRISPR /Trojan (CRIMIC), request from Lacin lab |
| Genetic reagent (*D. melanogaster*) | eve-p65.AD[attP2FRT2] | this study | | CRISPR /Trojan (CRIMIC), request from Lacin lab |
| Genetic reagent (*D. melanogaster*) | Fer3-GAL4-DBD | this study | | CRISPR /In frame insertion (C terminus), request from Lacin lab |
| Genetic reagent (*D. melanogaster*) | ems-GAL4-DBD | this study | | CRISPR /In frame insertion (C terminus), request from Lacin lab |
| Genetic reagent (*D. melanogaster*) | HLH4C-GAL4-DBD | this study | | CRISPR /In frame insertion (2nd exon), request from Lacin lab |
| Genetic reagent (*D. melanogaster*) | w1118;; fkh-T2A-GAL4-DBD/TM6b | this study | | CRISPR /In frame insertion (C terminus for RA isoform), request from Lacin lab |
| Genetic reagent (*D. melanogaster*) | w*;; D-VP16/TM6b | this study | | CRISPR /In frame insertion (C terminus), request from Lacin lab |
| Genetic reagent (*D. melanogaster*) | pJFRC29-10XUAS-IVS-myr::GFP-p10 in attP40 or attP2 | Rubin Lab | | |
| Genetic reagent (*D. melanogaster*) | pJFRC105-10XUAS-IVS-nlstdTomato in VK0003 | Rubin Lab | | |
| Genetic reagent (*D. melanogater*) | pJFRC12-10XUAS-IVS-myr::GFP attp40 or attP2 | Rubin Lab | | |
| Genetic reagent (*D. melanogater*) | pJFRC28-10XUAS-IVS-GFP-p10 in attP2 | Rubin Lab | | |
| Chemical compound, drug | Paraformaldehyde | EMS | 15713 | |
| Chemical compound, drug | Vectashield | Vectorlabs | H-1000 | |
| Chemical compound, drug | DPX | Electron Microscopy Sciences | 50980370 | |
| Chemical compound, drug | Gibson Assembly Master Mix | New England Biolabs | E2621S | |
| Recombinant DNA reagent | pCFD4-U6:1_U6:3tandemgRNAs | Addgene | 49411 | |
| Recombinant DNA reagent | pBS-KS-attB2-SA(1)-T2A-Gal4-Hsp70 | Addgene | 62897 | |

*Appendix 1 Continued on next page*

*Appendix 1 Continued*

| Reagent type (species) or resource | Designation | Source or reference | Identifiers | Additional information |
|---|---|---|---|---|
| Recombinant DNA reagent | pBS-KS-attB2-SA(1)-T2A-Gal4DBD-Hsp70 | Addgene | 62903 | |
| Recombinant DNA reagent | pBS-KS-attB2-SA(1)-T2A-p65AD-Hsp70 | Addgene | 62914 | |
| Recombinant DNA reagent | pBS-KS-attB2-SA(0)-T2A-Gal4-Hsp70 | Addgene | 62896 | |
| Recombinant DNA reagent | pBS-KS-attB2-SA(0)-T2A-Gal4DBD-Hsp70 | Addgene | 62902 | |
| Recombinant DNA reagent | pBS-KS-attB2-SA(0)-T2A-p65AD-Hsp70 | Addgene | 62912 | |
| Recombinant DNA reagent | pBS-KS-attP2FRT2-SA(0)-T2A-p65AD-Hsp70 | this study | | request from Lacin lab |
| Recombinant DNA reagent | pBS-KS-attP2FRT2-SA(1)-T2A-p65AD-Hsp70 | this study | | request from Lacin lab |
| Recombinant DNA reagent | pBS-KS-attP2FRT2-SA(2)-T2A-p65AD-Hsp70 | this study | | request from Lacin lab |
| Recombinant DNA reagent | pBS-KS-attP2FRT2-SA(0)-T2A-gal4DBD-Hsp70 | this study | | request from Lacin lab |
| Recombinant DNA reagent | pBS-KS-attP2FRT2-SA(1)-T2A-gal4DBD-Hsp70 | this study | | request from Lacin lab |
| Recombinant DNA reagent | pBS-KS-attP2FRT2-SA(2)-T2A-gal4DBD-Hsp70 | this study | | request from Lacin lab |
| Recombinant DNA reagent | pCFD4-exex | this study | | request from Lacin lab |
| Recombinant DNA reagent | pCFD4-vg | this study | | request from Lacin lab |
| Recombinant DNA reagent | pCFD4-H15 | this study | | request from Lacin lab |
| Recombinant DNA reagent | pUC57_Hb9 | this study | | request from Lacin lab |
| Recombinant DNA reagent | pUC57_vg | this study | | request from Lacin lab |
| Recombinant DNA reagent | pUC57_H15 | this study | | request from Lacin lab |
| Recombinant DNA reagent | pUC57_gw_OK2_Scro | this study | | request from Lacin lab |
| Recombinant DNA reagent | pUC57_gw_OK2_Ets21C | this study | | request from Lacin lab |
| Recombinant DNA reagent | pUC57_gw_OK2_eve | this study | | request from Lacin lab |
| Recombinant DNA reagent | pUC57_gw_OK2_Fer3 | this study | | request from Lacin lab |
| Recombinant DNA reagent | pUC57_gw_OK2_ems | this study | | request from Lacin lab |
| Recombinant DNA reagent | pUC57_gw_OK2_HLH4C | this study | | request from Lacin lab |

