## [Editor Report · eLife Assessment]

This work presents an **important** genetic toolkit for *Drosophila* neurobiologists to access and manipulate neuronal lineages during development and adulthood. The evidence supporting the fidelity of this toolkit after revision is **compelling**. This work will interest *Drosophila* neurobiologists in general, and some of the genetic tools may be used outside the nervous system. The conceptual approaches used in this paper are likely transferable to other fields as comparable data and genomic methods are obtained.

---

## [Referee Report · Reviewer #1 (Public review)]

The ventral nerve cord (VNC) of organisms like *Drosophila* is an invaluable model for studying neural development and organisation in more complex organisms. Its well-defined structure allows researchers to investigate how neurons develop, differentiate, and organise into functional circuits. As a critical central nervous system component, the VNC plays a key role in controlling motor functions, reflexes, and sensory integration.

Particularly relevant to this work, the VNC provides a unique opportunity to explore neuronal hemilineages-groups of neurons that share molecular, genetic, and functional identities. Understanding these hemilineages is crucial for elucidating how neurons cooperate to form specialized circuits, essential for comprehending normal brain function and dysfunction.

A significant challenge in the field has been the lack of developmentally stable, hemilineage-specific driver lines that enable precise tracking and measurement of individual VNC hemilineages. The authors address this need by generating and validating a comprehensive, lineage-specific split-GAL4 driver library.

Strengths and weaknesses:

The authors select new marker genes for hemilineages from previously published single-cell data of the VNC. They generate and validate specific and temporally stable lines for almost all the hemilineages in the VNC. They successfully achieved their aims, and their results support their conclusions. This will be a valuable resource for investigating neural circuit formation and function.

Comments on revisions:

The manuscript has been amended, and the points raised by the reviewers have been addressed.

---

## [Referee Report · Reviewer #2 (Public review)]

It is my pleasure to review this manuscript from Stoffers, Lacin, and colleagues, in which they identify pairs of transcription factors unique to (almost) every ventral nerve cord hemilineage in *Drosophila* and use these pairs to create reagents to label and manipulate these cells. The advance is sold as largely technical-as a pipeline for identifying durably expressed transcription factor codes in postmitotic neurons from single cell RNAseq data, generating knock-in alleles in the relevant genes, using these to match transcriptional cell types to anatomic cell types, and then using the alleles as a genetic handle on the cells for downstream explication of their function. Yet I think the work is gorgeous in linking expression of genes that are causal for neuron-type-specific characteristics to the anatomic instantiations of those neurons. It is astounding that the authors are able to use their deep collective knowledge of hemilineage anatomy and gene expression to match 33 of 34 to transcriptional profiles. Together with other recent studies, this work drives a major course correction in developmental biology, away from empirically identified cell type "markers" (in *Drosophila* neuroscience, often genomic DNA fragments that contain enhancers found to be expressed in specific neurons at specific times), and towards methods in which the genes that generate neuronal type identity are actually used to study those neurons. Because the relationship between fate and form/function are built into the tools, I believe that this approach will be a trojan horse to integrate the fields of neural development and systems neuroscience.

Comments on revisions:

The authors have addressed my (minor) suggestions.

---

## [Referee Report · Reviewer #3 (Public review)]

Summary:

Soffers et al. developed a comprehensive genetic toolkit that enables researchers to access neuronal hemilineages during developmental and adult time points using scRNA-seq analysis to guide gene cassette exchange-based or CRISPR-based tool building. Currently, research groups studying neural circuit development are challenged with tying together findings in the development and mature circuit function of hemilineage related neurons. Here, authors leverage publicly available scRNA-seq datasets to inform the development of a split-Gal4 library that targets 32 of 34 hemilineages in development and adult stages. The authors demonstrated that the split-Gal4 library, or genetic toolkit, can be used to assess the functional roles, neurotransmitter identity, and morphological changes in targeted cells. The tools presented in this study should prove to be incredibly useful to *Drosophila* neurobiologists seeking to link neural developmental changes to circuit assembly and mature circuit function. Additionally, some hemilineages have more than one split-Gal4 combination that will be advantageous for studies seeking to disrupt associated upstream genes.

Strengths:

Informing genetic tool development with publicly available scRNA-seq datasets is a powerful approach to creating specific driver lines. Additionally, this approach can be easily replicated by other researchers looking to generate similar driver lines for more specific subpopulations of cells, as mentioned in the Discussion.

The unification of optogenetic stimulation data of 8B neurons and connectomic analysis of the Giant-Fiber-induced take-off circuit was an excellent example of the utility of this study. The link between hemilineage-specific functional assays and circuit assembly has been limited by insufficient genetic tools. The tools and data present in this study will help better understand how collections of hemilineages develop in a genetically constrained manner to form circuits amongst each other selectively.

---

## [Author Response]

The following is the authors’ response to the original reviews

**Reviewer#1 (Public review):**
The ventral nerve cord (VNC) of organisms like *Drosophila* is an invaluable model for studying neural development and organisation in more complex organisms. Its well-defined structure allows researchers to investigate how neurons develop, differentiate, and organise into functional circuits. As a critical central nervous system component, the VNC plays a key role in controlling motor functions, reflexes, and sensory integration.Particularly relevant to this work, the VNC provides a unique opportunity to explore neuronal hemilineages - groups of neurons that share molecular, genetic, and functional identities. Understanding these hemilineages is crucial for elucidating how neurons cooperate to form specialized circuits, essential for comprehending normal brain function and dysfunction.A significant challenge in the field has been the lack of developmentally stable, hemilineage-specific driver lines that enable precise tracking and measurement of individual VNC hemilineages. The authors address this need by generating and validating a comprehensive, lineage-specific split-GAL4 driver library.Strengths and weaknessesThe authors select new marker genes for hemilineages from previously published single-cell data of the VNC. They generate and validate specific and temporally stable lines for almost all the hemilineages in the VNC. They successfully achieved their aims, and their results support their conclusions. This will be a valuable resource for investigating neural circuit formation and function.

We thank the reviewer for her/his positive comments and time reviewing our manuscript. We are pleased that the reviewer recognized the value of our work in generating a comprehensive, lineage-specific split-GAL4 driver library for VNC hemilineages. We agree that this will be a critical resource for investigating neural circuit formation and function, and we are encouraged by the positive comments regarding the novelty and potential impact of our approach.

**Reviewer#1(Recommendationsfortheauthors):**
I have no suggestions for further experiments, data, or analyses. There are some grammatical errors and referencing issues throughout, but the editors will hopefully catch them.

We appreciate the reviewer’s comments regarding the grammatical errors and referencing issues and have carefully checked the revised manuscript.

**Reviewer#2 (Public review):**
It is my pleasure to review this manuscript from Soffers, Lacin, and colleagues, in which they identify pairs of transcription factors unique to (almost) every ventral nerve cord hemilineage in *Drosophila* and use these pairs to create reagents to label and manipulate these cells. The advance is sold as largely technical-as a pipeline for identifying durably expressed transcription factor codes in postmitotic neurons from single cell RNAseq data, generating knock-in alleles in the relevant genes, using these to match transcriptional cell types to anatomic cell types, and then using the alleles as a genetic handle on the cells for downstream explication of their function. Yet I think the work is gorgeous in linking the expression of genes that are causal for neuron-type-specific characteristics to the anatomic instantiations of those neurons. It is astounding that the authors are able to use their deep collective knowledge of hemilineage anatomy and gene expression to match 33 of 34 transcriptional profiles. Together with other recent studies, this work drives a major course correction in developmental biology, away from empirically identified cell type "markers" (in *Drosophila* neuroscience, often genomic DNA fragments that contain enhancers found to be expressed in specific neurons at specific times), and towards methods in which the genes that generate neuronal type identity are actually used to study those neurons. Because the relationship between fate and form/function is built into the tools, I believe that this approach will be a trojan horse to integrate the fields of neural development and systems neuroscience.

We thank the reviewer for their time reviewing our manuscript, generous compliments, and appreciation of the potential of our study to drive a major shift in developmental biology, moving away from traditional marker-based methods toward utilizing the genes that mark neuronal type identity in “omics” datasets. Much like the Trojan Horse, which, though initially a concealed and subtle tool, we hope that the strategy outlined here will have continued impact, as we and others plan to leverage future high-resolution and developmental series of scRNAseq datasets to generate driver lines to target neuronal cell types with uttermost precision.

**Reviewer#2(Recommendationsfortheauthors):**
Line 126-127: I'm not sure if it is true to say "most TFs in the CNS are expressed in a hemilineage-specific manner." As the authors haven't formally interrogated how different neuronal features relate to expression patterns of all ~600 *Drosophila* TFs, how about replacing "most" with "many?"

The reviewer makes an excellent point. Work by Lacin and colleagues has demonstrated via genetic studies that lineage-specific transcription factors that regulate the specification and differentiation of postembryonic neurons are stably expressed during development. This was documented for 15 transcription factors in Lacin et al., 2014, and our lab has identified additional examples since. When we refer to the stable expression of transcription factors, we refer to such transcription factors, not the complete set of ~600 transcription factors described to date. We have added this citation to clarify this statement and replaced p6 line 135 ”Most” by “Many”. We have also address this now in the introduction (p5 line 109-116). Of note, as we conducted this study, we found that is closer to be a rule than an exception that if a transcription factor acted cluster as marker, it was also stably expressed during development. Thus, a growing number of transcription factors is now documented to be stably expressed in a hemilineage-specific manner

Line 265: Typo? 334 should be 34?

We thank the reviewer for noting this type error. We have corrected this typographical error.

Line 522: Refs 56, 57 here related to chinmo, mamo, br-c don't show br-c or mamo mark temporal cohorts of postmitotic neurons. Consider adding PMID: 19883497, 18510932, and 31545163.

We thank the reviewer for pointing this out and have added these references that demonstrate that broad, Mamo and Chinmo mark temporal cohorts in the developing adult CNS (p17 line 535).

**Reviewer#3 (Public review):**
Soffers et al. developed a comprehensive genetic toolkit that enables researchers to access neuronal hemilineages during developmental and adult time points using scRNA-seq analysis to guide gene cassette exchange-based or CRISPR-based tool building. Currently, research groups studying neural circuit development are challenged with tying together findings in the development and mature circuit function of hemilineage-related neurons. Here, authors leverage publicly available scRNA-seq datasets to inform the development of a split-Gal4 library that targets 32 of 34 hemilineages in development and adult stages. The authors demonstrated that the split-Gal4 library, or genetic toolkit, can be used to assess the functional roles, neurotransmitter identity, and morphological changes in targeted cells. The tools presented in this study should prove to be incredibly useful to *Drosophila* neurobiologists seeking to link neural developmental changes to circuit assembly and mature circuit function. Additionally, some hemilineages have more than one split-Gal4 combination that will be advantageous for studies seeking to disrupt associated upstream genes.Strengths:Informing genetic tool development with publicly available scRNA-seq datasets is a powerful approach to creating specific driver lines. Additionally, this approach can be easily replicated by other researchers looking to generate similar driver lines for more specific subpopulations of cells, as mentioned in the Discussion.The unification of optogenetic stimulation data of 8B neurons and connectomic analysis of the Giant-Fiber-induced take-off circuit was an excellent example of the utility of this study. The link between hemilineage-specific functional assays and circuit assembly has been limited by insufficient genetic tools. The tools and data present in this study will help better understand how collections of hemilineages develop in a genetically constrained manner to form circuits amongst each other selectively.Weaknesses:Although cell position, morphology (to some extent), and gene expression are good markers to track cell identity across developmental time, there are genetic tools available that could have been used to permanently label cells that expressed genes of interest from birth, ensuring that the same cells are being tracked in fixed tissue images.Although gene activation is a good proxy for assaying neurochemical features, relying on whether neurochemical pathway genes are activated in a cell to determine its phenotype can be misleading given that the Trojan-Gal4 system commandeers the endogenous transcriptional regulation of a gene but not its post-transcriptional regulation. Therefore, neurochemical identity is best identified via protein detection. (strong language used in this section of the paper).The authors mainly rely on the intersectional expression of transcription factors to generate split-Gal4 lines and target hemilineages specifically. However, the Introduction (Lines 97-99) makes a notable point about how driver lines in the past, which have also predominantly relied on the regulatory sequences of transcription factors, lack the temporal stability to investigate hemilineages across time. This point seems to directly conflict with the argument made in the Results (Lines 126-127) that states that most transcription factors are stably expressed in hemilineage neurons that express them. It is generally known that transcription factors can be expressed stably or transiently depending on the context. It is unclear how using the genes of transcription factors in this study circumvents the issue of creating temporally stable driver lines.

We thank the reviewer for their time to thoroughly and carefully review our manuscript. We appreciate the reviewer’s comments on its strengths, and we to hope that this body of work will prove to be incredibly useful to *Drosophila* neurobiologists seeking to link neural developmental changes to circuit assembly and mature circuit function. Likewise, we also appreciate the reviews careful consideration of its weaknesses, as the reviewer raises valid points. We have addressed these in our revised manuscript and believe this has significantly improved our manuscript.

Weakness 1: Although cell position, morphology (to some extent), and gene expression are good markers to track cell identity across developmental time, there are genetic tools available that could have been used to permanently label cells that expressed genes of interest from birth, ensuring that the same cells are being tracked in fixed tissue images.

The reviewer is fully correct, and we are aware of techniques developed by the laboratories of U. Banerjee, T. Lee, and J. Truman that can make transient GAL4 expression permanent, such as G-TRACE and lineage filtering. A common feature of these techniques is that effector activity is permanent (FLP-mediated removal of the FRT-flanked stop codon preceding GFP in G-TRACE or LexA in lineage filtering) but not the GAL4 activity, which is needed to take advantage of the vast UAS based effector lines such as RNAi libraries. For example, the study of Harris et al., 2015 from the Truman lab beautifully showed the strength of this kind of approaches for labeling the hemilineages but their approach cannot be used for functional studies for the reasons mentioned above. Fly lines using these approaches already have several transgenes and require the addition of several more to be used for functional studies. Our approach requires only two transgenes and is compatible with all UAS lines. One additional advantage of the splitGAL4 combinations that we identify here is that they are inserted in genes that are stably expressed throughout larval and pupal development in postmitotic cells, such that they can be used for functional manipulations during development. We emphasized this point in the discussion on page 16 under the heading “Mapping and manipulating morphological outgrowth patterns of hemilineages during development”.

Weakness 2: Although gene activation is a good proxy for assaying neurochemical features, relying on whether neurochemical pathway genes are activated in a cell to determine its phenotype can be misleading given that the Trojan-Gal4 system commandeers the endogenous transcriptional regulation of a gene but not its post-transcriptional regulation. Therefore, neurochemical identity is best identified via protein detection. (strong language used in this section of the paper).

We thank the reviewer for bringing up this important point. We agree that the Trojan-GAL4 approach will not faithfully recapitulate expression of genes that undergo posttranscriptional regulation. Our previous eLife paper (Lacin et al., 2019) showed that this is the case for Trojan driver lines for the ChAT gene. This study demonstrated that ChAT drivers unexpectedly but strongly labeled many GABAergic and Glutamatergic neurons in both the brain and VNC. With RNA in situ hybridization and immunostainings approaches, we showed that these neurons indeed express ChAT mRNA but not the protein. After our publication, another group showed a class of miRNA binds to the 3’UTR of the ChAT gene and regulates its expression post-transcriptionally (Griffith 2023). We believe that one major reason the Trojan driver lines do not faithfully recapitulate this expression pattern is due to the presence of the Hsp70 transcriptional terminator located at the 5’ end of the trojan exon which prematurely ends the transcript and affects the host gene’s 3’ UTR regulation. For this reason, we have recently generated new Trojan plasmids which allow the retention of the 3’UTR of the host gene in the transcript. We have revised the result section “Neurotransmitter use on pages 11-12 to address this point and have modified the language.

Weakness 3: The authors mainly rely on the intersectional expression of transcription factors to generate split-Gal4 lines and target hemilineages specifically. However, the Introduction (Lines 97-99) makes a notable point about how driver lines in the past, which have also predominantly relied on the regulatory sequences of transcription factors, lack the temporal stability to investigate hemilineages across time. This point seems to directly conflict with the argument made in the Results (Lines 126-127) that states that most transcription factors are stably expressed in hemilineage neurons that express them. It is generally known that transcription factors can be expressed stably or transiently depending on the context. It is unclear how using the genes of transcription factors in this study circumvents the issue of creating temporally stable driver lines.

We thank the reviewer for pointing out this apparent paradox, which we have clarified in the manuscript (p4. lines 94-102). Driver lines in the past have relied on the intersection of genes to label a defined set of neurons, which helped marking more narrow cell populations compared to enhancer traps in the adult CNS. Elegant and elaborate screening methods have been devised to identify hemidriver combinations that mark specific subset of neurons in the adult (Meissner et al, 2025 (eLife 98405.2) and citations therein). However, these hemidrivers do not leverage the expression pattern of hemilineage marker genes. Instead, their expression is controlled by random 2-3 kb genomic fragments. We and others observed that these drivers are not stably expressed during development. Hence, hemidrivers combinations that work beautifully to target adult neuronal cel populations can oftentimes not be directly used for developmental studies. Work by Lacin et al. 2014 has demonstrated that transcription factors that mark hemilineages are oftentimes stably expressed in the embryo larvae and even adult. When we made driver lines for these TF, using artificial exons, its complete endogenous enhancers elements remain intact. Consequently, we find that Trojan driver lines recapitulate the expression pattern of the transcription factor gene in which it was inserted, and the hemidrivers are stably expressed during development. Hence, leveraging scRNAseq cluster markers for hemilineages and converting them to Trojan driver lines, the approach we took in this paper, has proven a powerful method to generate stable driver lines for developmental studies.

**Reviewer#3(Recommendationsfortheauthors):**
(1) Line 14: Affiliations typo should be correct to "St. Louis".

We thank the reviewer for catching this and have corrected the typo.

(2) Line 26: "model systems have focused on only on a few".

We have replaced the words “a few regions” by “select regions” to better contrast that studies to date have been performed, but not at CNS level, due to the lack of genetic driver lines.

(3) Line 52: The use of "medium" here is ambiguous without a comparison.

We agree that the term “medium” in line 52 could be ambiguous without context, and we appreciate your suggestion to clarify this. The revised sentence now reads: “*Drosophila* has served as a powerful model system to investigate how neuronal circuits function due to its medium complexity compared to vertebrate models”

(4) Line 91-92: Consider shortening to "of behavioral circuit assembly".

Thank you for this suggestion, we have revised p4 lines 90-91 to: “Thus, taking a hemilineage-based approach is essential for a systematic and comprehensive understanding of behavioral circuit assembly during development in complex nervous systems.”

(5) Line 216-217: Consider establishing what the expected morphology and neurochemical phenotype for 2A neurons is before presenting findings.

This suggestion is well-taken, and agree that this paragraph did not fully get the point across we were trying to make. This purpose of this paragraph is to explain our workflow of how we assigned 16 hemilineages to orphan clusters, which is why we present the data in this order and present the morphology of hemilineage 2A last. To accommodate the reviewer’s suggestion, we have now clarified our approach before diving into the results to improve the flow of this paragraph (p8 lines 218-223). Briefly, the starting point to annotate the 16 orphan scRNAseq clusters was each time taking one orphan scRNAseq cluster, picking its top cluster marker genes that had not been established yet as marker genes for any hemilineage, and visualizing the morphology of the neurons that expressed such cluster marker using a reporter line for the cluster marker or an antibody stain for its protein. We then compared this to documented hemilineage morphologies, and to narrow down our search, we compared the observed trajectories to those of unannotated hemilineages that used the same neurotransmitter as the orphan scRNAseq. The evaluation of the documented morphologies of the hemilineages came at the last part of our method to annotate the hemilineages to orphan scRNAseq clusters, which is why we chose to present the expected morphology of a hemilineage at the end.

(6) If "neurochemical" phenotype and "neurotransmitter" identity are sometimes used interchangeably but seem to mean the same thing. Consider choosing one term throughout.

We thank the reviewer for this suggestion and have changed the terminology to “neurotransmitter use” (p11-12 lines 326-359).

(7) Line 235: MARCM technique citation needed.

We thank the reviewer for pointing this out, the citation (no. 37, p9 line 249) was present in the method section, but we had inadvertently omitted it in the main text and we have now corrected this.

(8) Line 281: typo, should be "patterns".

We thank the reviewer for noting this and have corrected this.

(9) Line 469: End of sentence needs a ".".

We have added the punctuation mark.

(10) Line 516: "driver line combinations to express...".

We have inserted the word “to” to correct it.

(11) Please make sure that the correct genotypes are matched in the figure legends and Table 1. For instance, knot-GAL4-DBD is listed as the hemi driver for 10B neurons in Figure 3 but only knot-p65.AD is listed in Table 1.

We thank the reviewer for catching this, we made a mistake and the correct hemidriver combination used in Figure 3L i: knot-GAL4-AD with hb9-GAL4-DBD. We have updated the legend and carefully checked the legends and tables.

(12) Consider making different color choices for readability when possible and be consistent with labeling CadN. For instance, in Figure 1 the magenta color has three separate meanings: CadN, Acj6, and unc-4. Either of the three genes can be mistaken for the other for a reader mainly paying attention to the magenta color. I find that one color can mean two things in a figure if organized properly but any more begs for confusion. Also, CadN can be easily labeled if used in a new figure (e.g. Figure 1-Supplment 1).

We thank the reviewer for this insightful observation and have adjusted figure 1 so that cadN is displayed in blue and reporter genes expressing Acj6, Unc-4 or their intersection in green. The legend is modified to reflect these changes.

(13) If Seurat object changes or additional quality control steps were taken from the original studies, please provide these changes. Similarly, provide any scRNA-seq code used or cite code used for readers to access. Also, provide a section in the methods briefly describing how genes were chosen (criteria) for tool development.

We thank the reviewer for nothing we had not described our scRNA analysis pipeline and criteria to select transcription factors in the methods section of the manuscript. We have added this section at p19 lines 548-558. Briefly, we used the Seurat object generated by Allen et al., 2015, and did not change quality control steps, normalizations or scaling. Candidate genes to make split-GAL4 drivers from were chosen based on their ability to mark the clusters defined by Allen et al. We did not use computer-based algorithms and made a list of the top cluster markers. Then, we made binary combinations amongst these cluster markers and with hemilineages markers we had identified before (Lacin et al, 2014; Lacin et al 2019), and used the code generated by Allen et al., 2015 (deposited on Github) with Seurat v5 to test if these combinations marked unique clusters. We then prioritized testing these combinations based on the availability of antibodies, BAC lines and CRiMIC/MiMIC constructs to validate their expression pattern prior to creating split-GAL4 lines for these candidates.

(14) In regard to the seemingly contradictory argument that most transcription factors are stably expressed when most drivers of the past used regulatory elements of transcription factors: the paper could be strengthened by either (a) describing how older driver lines differ from the lines presented in the paper or (b) remarking on the endogenous temporal stability of the transcription factors used in this study.

We thank the reviewer for pointing this out, and we agree that it is necessary to clarify this apparent paradox since it is essential for understanding the impact of the present work. We have revised our manuscript described in our response to weakness 1.